

# NO₂ vertical profiles and column densities from MAX-DOAS measurements in Mexico City

Martina Michaela Friedrich[1,2], Claudia Rivera[1,3], Wolfgang Stremme[1], Zuleica Ojeda[1], Josué Arellano[1,4], Alejandro Bezanilla[1], José Agustín García-Reynoso[1], and Michel Grutter[1]

[1]Centro de Ciencias de la Atmósfera, Universidad Nacional Autónoma de México
[2]Belgian Institute for Space Aeronomie (BIRA-IASB)
[3]Facultad de Química, Universidad Nacional Autónoma de México
[4]Instituto de Geofísica, Universidad Nacional Autónoma de México

**Correspondence:** M.M. Friedrich (martina@atmosfera.unam.mx)

**Abstract.** We present a new numerical code, Mexican Maxdoas Fit (MMF), developed to retrieve profiles of different trace gases from the network of MAX-DOAS instruments operated in Mexico City. MMF uses differential slant column densities (dSCDs) retrieved with the QDOAS (Danckaert et al., 2013) software. The retrieval is comprised of two steps, an aerosol retrieval and the trace gas retrieval that uses the retrieved aerosol profile in the forward model for the trace gas. For forward

model simulations, VLIDORT is used (e.g. Spurr et al., 2001; Spurr, 2006, 2013). Both steps use constrained least square fitting, but the aerosol retrieval uses Tikhonov regularization and the trace gas retrieval optimal estimation. Aerosol optical depth and scattering properties from the AERONET database, averaged ceilometer data, WRF-Chem model data as well as temperature and pressure sounding data are used for different steps in the retrieval chain.

The MMF code was applied to retrieve NO₂ profiles with two degrees of freedom (DOFs=2) from spectra of the MAX-DOAS

instrument located at the UNAM campus. We describe the full error analysis of the retrievals and include a sensitivity exercise to quantify the contribution of the uncertainties in the aerosol extinction profiles to the total error. A dataset comprised of measurements from January 2015 to July 2016 was processed and the results compared to independent surface measurements. We concentrate on the analysis of 4 single days and additionally present diurnal and annual variabilities from averaging the 1.5 years of data. Even though the total error is considerably large (depending on the exact counting 14 –20 %) this work still

provides new and relevant information about NO₂ in the boundary layer of Mexico City.

## 1   Introduction

Air pollution is a serious environmental problem due to its negative impacts on human health and ecosystems. Fast growing urban and industrial centers are continuously affected by bad air quality and in order to assess their current efforts to mitigate emissions and plan for more efficient strategies to lower the concentration levels of harmful contaminants, it is indispensable

to have the proper tools to measure them not only at ground level but also throughout the boundary layer. The MAX-DOAS technique (e.g. Hönninger et al., 2004; Platt and Stutz, 2008) has rapidly developed in recent years and has proven extremely





valuable in tropospheric chemistry and air pollution studies, since it provides vertical distribution of trace gases with high temporal resolution.

This remote sensing technique is based on the spectroscopically resolved measurement of scattered sunlight at different elevation angles, allowing for the retrieval of total column amounts of aerosols and trace gases with profiling capability. Powerful applications of this technique have demonstrated to provide useful information about the vertical distribution of aerosols (Frieß et al., 2006; Wang et al., 2016) and photochemically relevant species such as nitrogen dioxide ($NO_2$), formaldehyde (HCHO), glyoxal (CHOCHO) and nitrous acid (HONO) among other gases (e.g. Wittrock et al., 2004; Wagner et al., 2011; Ortega et al., 2015; Hendrick et al., 2014).

Photochemical reactions involving $NO_2$ play an important role in the formation of $O_3$ (Finlayson-Pitts and Pitts, 2000). The Mexico City Metropolitan Area (MCMA) has been particularly affected since the 1990's by high $O_3$ episodes threatening the population and forcing the authorities to impose strict restrictions in the usage of motor vehicles (Molina and Molina, 2002). Measurements have been performed in the region using fixed and mobile DOAS zenith-scattered sunlight (Melamed et al., 2009; Johansson et al., 2009; Rivera et al., 2013) and in 2014 a MAX-DOAS network, initially consisting of four instruments, was established in the Mexico City Metropolitan Area (MCMA) and has been operating since (Arellano et al., 2016).

In this contribution, we describe the MMF (Mexican Maxdoas Fit) code that has been implemented to retrieve vertical distribution of aerosols and trace gases giving emphasis on the errors and diagnostics of the results. An overview of the MAX-DOAS instruments is provided in Section 2 and the complete retrieval strategy from the measured spectra to vertical trace gas profiles is summarized in Fig. 1.

Radiative transfer simulations (constituting the forward model, yellow boxes in Fig.1) are performed with VLIDORT (Spurr, 2013) to derive simulated differential slant column densities (dSCDs) at the middle of the corresponding wavelength interval used to derive dSCDs from measured spectra. The dSCD retrieval (blue boxes in Fig. 1) is performed with QDOAS (Danckaert et al., 2013) and is described briefly in Sect.3. The orange parts in the figure refer to the aerosol retrieval while the green parts belong to the trace gas retrieval. Details on the forward model choice, the forward model input calculation (light blue box in Fig. 1) and processing of output quantities in the inversion algorithm are described in Sect.4.

An error analysis has been included in Sect. 5, especially investigating the effect of the aerosol retrieval on the $NO_2$ results. Some examples of the $NO_2$ variability are provided from one of the network's stations and compared to surface concentrations in Sect.6 and a summary of the work and an outlook on planned improvements to the retrieval code is presented in Sect. 7.

## 2 Instruments

An instrument based on the MAX-DOAS technique was designed and developed by the Center for Atmospheric Sciences at UNAM. It consists of two main parts: the scanner unit which collects the scattered light, and the acquisition/control unit, containing a spectrometer and computer that records and stores the measurements. The two components are connected by an optical fiber and a data connection cable. Both are described briefly in the following sections. For a more complete description we refer the reader to Arellano et al. (2016).





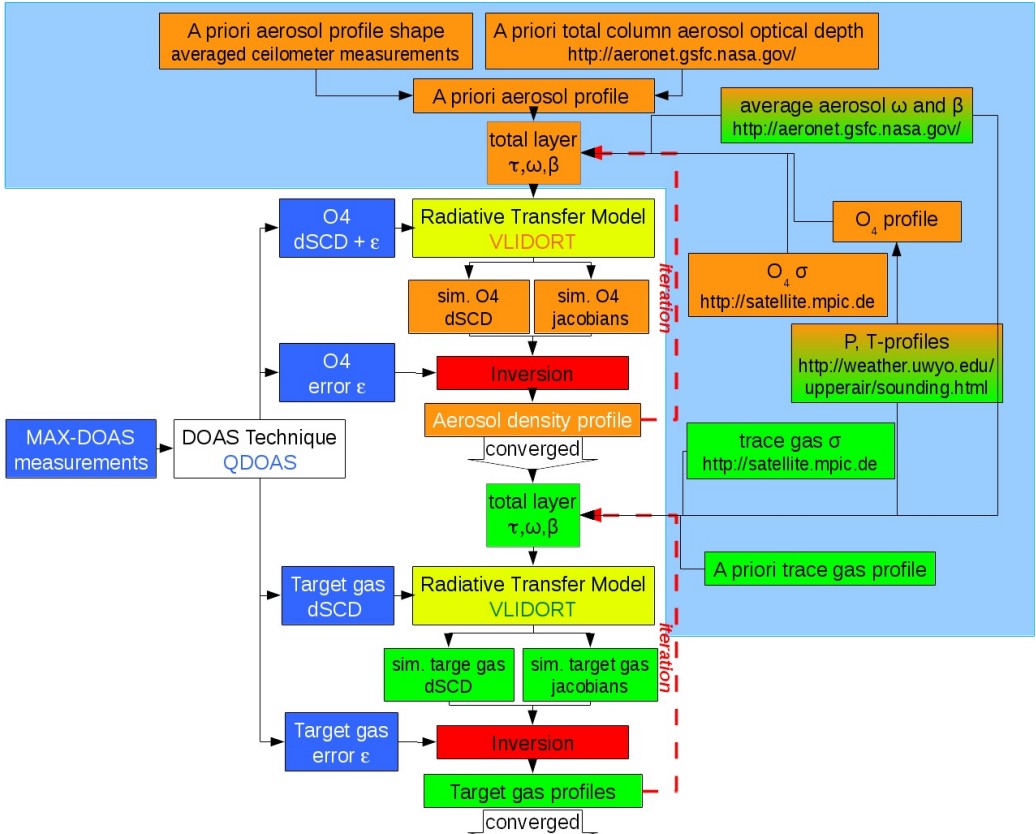

**Figure 1.** Flowchart of the complete trace gas retrieval algorithm. Orange boxes belong to the aerosol retrieval, green boxes to trace gas retrieval. The light blue box encompasses forward-model-input calculation. The dark blue boxes are in/ outputs of QDOAS.

## 2.1 The scanner

The scanner is composed of a plastic enclosure (NEMA-rated type 3) resistant to sunlight and hermetically sealed to protect the inner parts from water and bugs. This is important in order to assure large measurement periods under harsh conditions (Hönninger et al., 2004; Galle et al., 2010; Arellano et al., 2016). The scattered light is collected by a a plano-convex quartz

5 lens (Edmund Optics, ⌀ = 25.4 mm, $f$ = 100 mm), focusing it into the entrance of an optical fiber (Fiber Tech Optica, quartz, 6 m long, ⌀ = 0.6 mm) which transfers the light into the acquisition/control unit. These optical parts are mounted inside a telescope housing constructed of Nylamid material. A shutter system is installed also within the telescope. It consists of a stepper motor (Mercury, 7.5° by step ) that rotates a metal circular plate to prevent the passage of light into the optical fiber and an optical switch which is used to indicate the position of the plate. The shutter is used to make measurements of dark spectra

10 in between scans.



A stepper motor (Oriental Motors, PK266-02A) inside the enclosure allows the movement of the scanner unit in a range of 180° with steps of 0.1°. A mechanical switch is used as a reference to indicate the starting position of each scan. The motor is controlled by an electronic board composed of an 8-bit microcontroller (AVR- architecture), a RS-232 port for communication with the acquisition/control unit, a temperature sensor (Maxim 18B20, accuracy ±0.5°C), and a dual axis accelerometer (Ana-

log Devices, accuracy ±0.1°) that provides an accurate determination of the telescope's pointing elevation. The theoretical field of view (FOV) of the optical system is 0.31°.

## 2.2    The acquisition/control unit

The second part of the instrument consists of a metallic housing receiveing the collected light through the optical fiber and sending it to the spectrometer (Ocean Optics, USB2000+). This commercial device has a crossed asymmetric Czerny-Turner

configuration, diffraction grating (1800 lines/mm) and a slit size of 50 $\mu$m wide x 1 mm high, recording spectra in a wavelength range of 289 - 510 nm at a resolution of 0.69 nm (full width at half maximum). It uses a Charge-Coupled Device (CCD) detector array (Sony ILX511B) of 2048 pixels with an integration time adjustable between 1 ms to 65 s.

Because changes in temperature can affect the wavelength/pixel ratio and also modify the optical properties of the spectrometer like the alignment and the line shape (e.g. Carlson et al., 2010; Coburn et al., 2011), a temperature control system

composed of a Peltier cell (Multicomp) and three temperature sensors (Maxim 18B20, accuracy ±0.5°C) controlled by an electronic board were implemented. The cooling side of the Peltier cell was placed on top of the spectrometer and the heating side was attached to a heat sink. The Peltier cell and the spectrometer were wrapped in a styrofoam box to keep the temperature insulated from the outside. The three temperature sensors were placed on the heat sink, the Peltier cell and spectrometer to monitor temperature changes. The temperature control system was wrapped in an aluminum enclosure to prevent the heat

spreading to other parts of the system and a fan was installed to extract the heat from the enclosure.

The electronic board for the temperature control is composed of an 8-bit microcontroller (AVR, architecture) and a RS-232 communication port. This board is responsible for obtaining the data from the sensors and adjusting the voltage in the Peltier cell to keep the temperature constant.

The acquisition/ control unit has a laptop computer (Dell, Latitude 2021) with Linux operating system contained within

the enclosure. The program controlling the hardware is written in C++ using Qt libraries. A script is used to carry out the measurement sequence previously defined and to monitor the spectrometer temperature.

## 2.3    Measurement strategy

A complete scan consists of a sequence that begins with a measurement towards the zenith, (90 ° elevation angle), followed by four measurements between elevation angles 0 ° and 10 ° towards the west (for UNAM station, this is 85 ° azimuth angle),

then 16 measurements are taken with elevation angles between 10 ° elevation angle towards west and 10 ° elevation angle towards the east (crossing the zenith, but without taking a measurement), followed by four measurements between 10 ° and 0 ° elevation angle towards the east. The same sequence is then repeated but in reverse order. At the end of this cycle, a dark spectrum with the closed shutter is taken. With this setup, a complete scan takes about 7 minutes. The measurement sequence



is likely to change in the future to use longer integration times but at the same time reduce the number of viewing directions in the range between $10\,°$ elevation angle towards the west and $10\,°$ elevation angle towards the east in order to keep the total scan time roughly constant. All data presented in this manuscript uses this setup meaning that we include both westerly and easterly viewing directions in the same retrieval and hence the result represents and average. This strategy leads to larger fitting errors

since the assumption of horizontal inhomogenety is likely to be less true. With our retrieval chain, it is possible to consider the easterly and westerly directions separately to investigate the differences in viewing directions which is subject of current investigation.

The output from the MAX-DOAS instruments consists of five files per complete scan: A file containing the meta data of each spectrum (e.g. accelerometer data for each measurement, time of the acquisition, temperatures within the acquisition unit), all

the spectra in non-zenith directions, the dark spectrum measured with the shutter closed, a meta file for the dark spectrum and the first zenith reference measurement. These files are stored for further processing.

## 3   Differential Slant Column Densities (dSCDs) Retrieval

The spectra were evaluated using the QDOAS (version 2.105) software (Danckaert et al., 2013). As a pre-processing step before the QDOAS analysis, the dark signal was subtracted from each of the measurement spectra. A wavelength calibration

was conducted in QDOAS by applying a non-linear least squares fit to a solar atlas (Kurucz et al., 1984).

For $NO_2$, the retrieval was conducted in the 405 to 465 nm wavelength range using the spectrum measured at the zenith position at the beginning of each of the measurement sequences as reference. For the analysis, differential cross-sections of $NO_2$ at 298 K (Vandaele et al., 1998), $O_3$ at 221 K and 241 K (Burrows et al., 1999) and the oxygen dimer (Hermans et al., 1999) were convolved with the slit function of the spectrometer and a wavelength calibration file (created using a mercury

lamp) and using the convolution tool of the QDOAS software. A Ring spectrum, generated at 273 K from a high resolution Kurucz file using the Ring tool of the QDOAS software (Danckaert et al., 2013), was also included in the analysis.

For $O_4$, the retrieval was conducted in the 336 to 390 nm wavelength range. Differential cross-sections of $O_4$ (Hermans et al., 1999), $O_3$ at 221 K and 241 K (Burrows et al., 1999), $NO_2$ at 294 K (Vandaele et al., 1998), BrO at 298 K (Wilmouth et al., 1999), HCHO at 298 K (Meller and Moortgat, 2000) and a Ring spectrum were included in the analysis. A 3rd degree

polynomial was used for the retrievals.

Figure 2 shows the dSCDs retrieval error statistics for $NO_2$ as a function of elevation angle. The plot shows results of 31,448 scans from January to December 2016. Larger dSCDs fitting errors are found for viewing angles closer to the horizon ( smaller elevation angles), likely due to physical interferences during the measurements. As elevation angles approach the zenith, retrieval dSCDs errors decrease considerably.





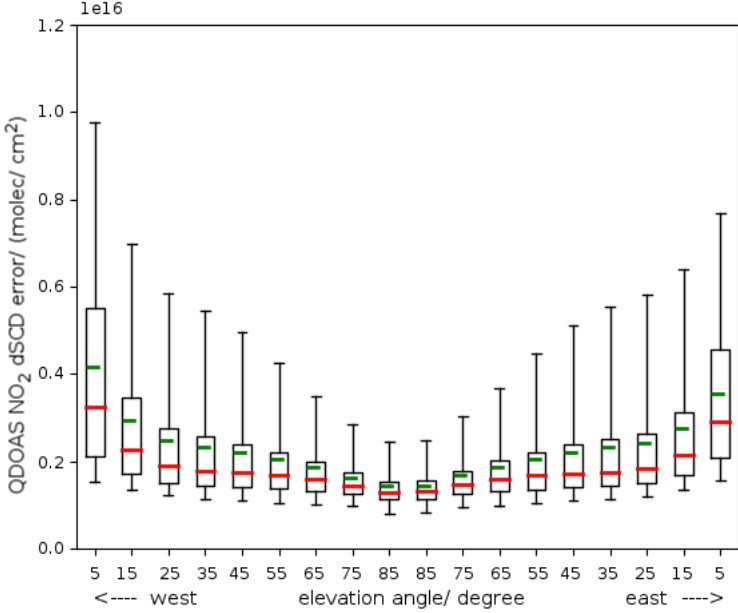

**Figure 2.** dSCDs measurement error statistics for NO$_2$ at the UNAM station as a function of elevation angle for data taken in the year 2016. The box encloses the 25-75 percentile, the whiskers are 5-95 percent, the green bar is the mean and the red bar the median.

## 4  Mexican Maxdoas Fit

The method for the trace gas retrieval from slant column densities using the MMF code is comprised of two parts, an aerosol retrieval using the known O$_4$ profile, and the trace gas retrieval (e.g. Platt and Stutz, 2008). Both parts consist of the same steps: A foward model and an inversion algorithm.

5    In Sect.4.1, details about the inversion strategy are given. Our choice of forward model, VLIDORT (e.g. Spurr et al., 2001; Spurr, 2006, 2013), and the input parameter calculation are detailed in Section 4.2. How forward model output quantities are processed for the inversion step is detailed in Sect.4.3.

   MMF has been participating in the Round-Robin comparison of different retrieval codes for the FRM4DOAS project (Frieß et al., in preparation). It has also participated in the profile retrieval from dSCD from the CINDI2 campaign, both for NO$_2$ and

10  HCHO (Tirpitz et al., in preparation) as well as for HONO (Wang et al., in preparation). The retrieval time per aerosol and trace gas retrieval with the Mexico City set-up is roughly half a minute for each scan, but highly dependent on the conditions.

### 4.1  Inversion theory

The inversion strategy relies on the fact that the problem is not too non-linear so that in the iteration procedure, the new value for the quantity vector in question $x$ (either the aerosol total extinction per layer or the trace gas optical depth per layer) can be




calculated using a Gauss-Newton (GN) scheme according to Eq. 1 (Rodgers, 2000). This step corresponds to the red box and arrows in Fig. 1.

$$x_{i+1} = x_a + (K_i^T S_m^{-1} K_i + R)^{-1} K_i^T S_m [(y - F(x_i)) - K_i(x_a - x_i)] \tag{1}$$

In a recent update of the code, implemented after the analysis presented here (i.e. not used for obtaining the results here) this
GN iteration scheme was replaced by a slightly slower[1] but more stable Levenberg Marquardt (LM) iteration scheme in order
to enable working in the logarithmic retrieval space which makes the problem more unlinear: Eq. 1 was replaced by Eq.2 for
more non-linear inversion problems (Rodgers, 2000)

$$x_{i+1} = x_i + \left[(1+\gamma)R + K_i^T S_m^{-1} K_i\right]^{-1} \left[K_i^T S_m^{-1}(y - F(x_i)) - R(x_i - x_a)\right] \tag{2}$$

In both equations, superscript $T$ denotes transposed, superscript $-1$ denotes the inverse. The index $i$ is the iteration index,
the subscript $a$ indicates a-priori values. $S_m$ is the measurement error covariance matrix, $y$ denotes the vector of measured
differential slant column densities. $F(x_i)$ are the simulated differential slant column densities, calculated using the forward
model with input profile $x_i$. Both $y$ and $F(x_i)$ are vectors of dimension (# telescope viewing angles). $K_i = \partial F(x)^l / \partial x^n$ is the
jacobian matrix at the $i$-th iteration describing the change of simulated dSCD for viewing angle $l$ when the profile $x$ in layer $n$
is varied.

In case of the use of Eq.2, the new $x_{i+1}$ is only accepted if the cost function in Eq. 3 decreases w.r.t the previous cost-function

$$\text{cost} = \sum_k^{\text{angles}} \sum_j^{\text{angles}} (y - F(x))_k S_m^{-1\,kj}(y - F(x))_j + \sum_k^{\text{layers}} \sum_j^{\text{layers}} (x - x_a)_k Sa^{-1\,kj}(x - x_a)_j \tag{3}$$

If this is the case and $(1+\gamma)$ is not yet equals to 1, the factor $(1+\gamma)$ is halved for the next iteration. If however, the cost-function increases, the newly calculated $x_{i+1}$ is discarded and the i-th calculation repeated with a factor $(1+\gamma)$ increased by a factor of 16.

In this study, the GN iteration scheme was used and the retrieval grid equals the simulation grid. Details about the layer distribution are given in Sect. 4.2.

In the case of optimal estimation (OE), the regularization matrix $R$ is equal to the inverse of the a-priori covariance matrix, $R = S_a^{-1}$. OE regularization is used for trace gas retrieval. The LM scheme of Eq. 2 has currently only been tested with this choice of regularization matrix. Other regularization matrices are possible, see e.g. Steck (2002).

[1]In order to counteract the slowdown of the retrieval, more restrictions were placed on the observation geometry for a single scan: a single relative azimuth angle and a single solar zenith angle per scan. This means in particular that two different viewing directions cannot be treated as a single scan any longer. Although this means a significant cut in flexibility, it results in a retrieval time speed up of a factor of 4 and a more typical retrieval time per scan is around 5 seconds.





For the aerosol retrieval used in this study, we use the $L1$ operator ($R = L1^T \alpha L1$) where the scaling parameter $\alpha$ is supplied via an input script to limit the degrees of freedom (DOF) to just slightly above 1. Different scalings for the upper layers and lower layers can be supplied, as well as a complete regularization matrix $R$.

Tests using the logarithm of the partial layer vertical column density (for $NO_2$ retrieval) or layer extinction profile (for aerosol
retrieval) motivated the change to the LM iteration scheme due to the increased non-linearity when working in a semi-log space as state-measurement space. The studies performed during the FRM4DOAS Round-Robin analysis of synthetic data (Frieß et al., in preparation) and the CINDI2 retrieval exercises for HCHO and $NO_2$ (Tirpitz et al., in preparation) and IO (Wang et al., in preparation) used the retrieval in logarithmic space with LM iteration scheme.

## 4.2 Forward model

Several radiative transfer codes have been developed to serve as forward models for this kind of retrieval algorithms. There are Monte Carlo (MC) codes, such as AMFTRAN (Marquard, 1998), TRACY (von Friedeburg et al., 2002; von Friedeburg, 2003) or PROMSAR (Palazzi et al., 2005). All these MC radiative transfer codes start with ejecting photons from the instrument and following the photon path backwards (see also e.g. Perliski and Solomon, 1993; Marquard et al., 2000), hence they are sometimes somewhat confusingly referred to as backward models. The advantage of MC codes is their high accuracy, the
disadvantage is the rather long calculation time.

Another class of radiative transfer codes, such as VLIDORT and DISORT (Stamnes et al., 1988; Dahlback and Stamnes, 1991), uses the discrete ordinate algorithm. This finite difference method is based on finding solutions for an atmosphere consisting of a number of homogeneous layers using gaussian quadrature approximations to the integro-differential radiative transfer equation expressed using Legendre polynomials for the phase function and Fourier series for the intensity.

The big advantage of the second class of codes is their heigh speed. Another advantage of VLIDORT is that it calculates not only the intensity but also analytic Jacobians. Therefore, instead of (2 × # layers) calls to the program to perform a finite difference approximation, one call is sufficient. MMF uses the intensity part of VLIDORT, version 2.7, released in August 2014. (V)LIDORT is configured to return intensity jacobians w.r.t. gas absorption layer properties or aerosol total extinction layer properties.

Layers above the retrieval grid can be added for the forward model simulation. Both grids are supplied via an input file to the code together with the corresponding a-priori values of the trace gas concentration and the aerosol profile. In this study, the simulation grid is identical to the retrieval grid and the layer distribution of the retrieval grid is not equidistant. The grid consists of 22 layers up to 25 km. The layer thickness increases from 100 m at the lowest layer to 5 km in the upper-most layer. The exact height distribution can be seen in Fig. 5. The surface albedo used in this study was set to 0.07, this value can
be passed to MMF in the configuration file. However, in practice the effect for downwelling intensity and hence for the dSCD calculation was found to be negligible.

For each atmospheric input layer, the following input needs to be supplied to VLIDORT[2]:

---

[2]We only use the LIDORT part of VLIDORT, meaning that we only consider the total intensity and not the polarization




1. total layer optical depth $\tau$

2. single scattering albedo $\omega$

3. phase function expansion coefficients $\beta$

Since Jacobians with respect to changes of quantity $\chi$ in each layer are required, the normalized derivatives of the total

primary optical quantities, $\tau$, $\omega$ and $\beta$ with respect to this quantity need to be supplied as well. In case of trace gas retrieval, $\chi = a_{\mathrm{gas}}\Delta h$, and in case of aerosol retrieval, $\chi = (\sigma_{\mathrm{aer}} + a_{\mathrm{aer}})\Delta h$. Here, $a_{\mathrm{gas}}$ and $a_{\mathrm{aer}}$ are the gaseous and aerosol absorption coefficients, respectively and $\sigma_{\mathrm{aer}}$ is the aerosol scattering coefficient. $\Delta h$ is the layer thickness. Hence, for each layer the following input is additionally required:

4. rate of change of $\tau$ w.r.t. $\chi$

5. rate of change of $\omega$ w.r.t. $\chi$

6. rate of change of $\beta$ w.r.t. $\chi$

In order to calculate this input, we first need to calculate the separate properties for the trace gases and the aerosols. Afterwards, these quantities are combined to yield the total layer quantities. The part of the input calculation is inclosed in Fig. 1 in the light blue box.

The calculation of the contribution from the trace gas and the air density (through Rayleigh scattering) for the layer inputs (1-6) is presented in Sect.4.2.1. The aerosol contribution calculation is outlined in Sect.4.2.2. These two sections also contain information on the source of information for the respective a-priori values, as well as, for gas, the error covariance matrix.

The calculation of the contributions from gas and aerosol to yield the VLIDORT inputs (1-6) is detailed in Sect. 4.2.3.

### 4.2.1    Input for gases

In Fig. 3(a), the steps to calculate the contribution of the trace gas and the contribution from Rayleigh scattering in each layer is outlined. Red boxes represent the trace gas (and air) contribution to VLIDORT input quantities, green boxes are primary inputs to the inversion code, light blue boxes are intermediate quantities that are neither direct input to MMF, nor final input for the forward model.

The a-priori volume mixing ration (VMR) profile and the covariance matrix for this study are calculated from simulations
covering the year 2011 using WRF-Chem V3.6. The model domain covers Mexico and surrounding seas, the domain has 200 by 100 square grid cells with approximately 28 km width and 35 vertical layers. The parameterizations used were for Micro Physics WRF single moment 3-class (4), for PBL the Yonsei University scheme (1), for Cumulus the Grell 3D ensemble scheme (5), for radiation the Goddard shortwave scheme (1) and the Dudhia long wave scheme (4), time-varying sea surface temperature on, grid analysis nudging on, the unified Noah Land Use surface model (2), off an urban canopy model and the
2008 emissions inventory were used.





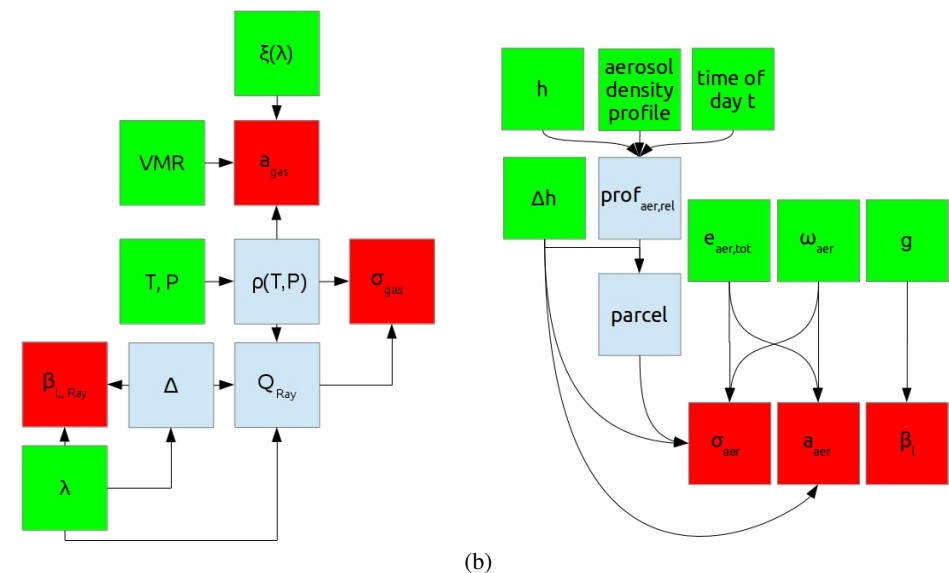

**Figure 3.** Calculating the gas and air (a) and aerosol (b) contribution to the layer input parameters for VLIDORT. The green boxes are primary input to MMF, blue boxes are intermediate quantities and red boxes final output that are combined for VLIDORT layer input. VMR denotes here the a-priori volume mixing ratio. Instead of a VMR, an a-priori trace gas density can be supplied, too.

MMF takes pressure and temperature profiles on a separate height grid, internal interpolation to the provided retrieval grid is performed. In the processing chain implemented in Mexico, the temperature and pressure profiles from radiosonde data for the specific day are downloaded from the University of Wyoming (http://weather.uwyo.edu/upperair/sounding.html) from the Mexico City Airport, station number 76679. These temperature and pressure profiles are used to calculate the dry air number density $\rho$, see Appendix A.

The absorption cross section $\xi_\lambda$ is taken at a wavelength in the middle of the wavelength interval used for the QDOAS retrieval (see Sect.3.), for aerosol retrieval it is 361 nm and for $NO_2$ it is 414 nm. The same cross-section tables as for QDOAS dSCD retrievals (see Sect. 3) are used for $NO_2$ and $O_4$. However, since the cross section cancels out at the end of the calculation, its exact value is not too important.

There are analytical fits to the temperature dependence, using e.g. linear (e.g. Vandaele et al., 2003, for $NO_2$) or more complicated (Kirmse et al., 1997, e.g.) temperature dependences. For a comparison study of different fitting functions, see Orphal (2002). These could be implemented, however, the cross-section is currently not temperature dependent, in agreement with the assumption made for defining dSCDs.

The trace gas absorption coefficient is calculated according to Eq.4.

$$a_{gas} = \text{VMR} \cdot \xi_\lambda(T) \cdot \rho \tag{4}$$



For the calculation of the depolarization ratio $\Delta$, the main contributions are from $N_2$, $CO_2$, $O_2$ and Ar. Our implementation follows Bates (1984). Details are given in Appendix A. For the calculation of the Rayleigh cross-section $Q_{\mathrm{Ray}}$ we follow the implementation of Goody and Yung (1989) (their equation 7.37), see also Platt et al. (2007) (their equation 19.5). The air scattering coefficient can then be calculated according to Eq. 5

$$\sigma_{\mathrm{air}} = Q_{\mathrm{Ray}} \cdot \rho \qquad (5)$$

and the Rayleigh scattering expansion coefficients $\beta_l$ (e.g. Spurr et al., 2001), according to Eq. 6

$$\beta_{\mathrm{air0}} = 1, \ \beta_{\mathrm{air},1} = 0, \ \beta_{\mathrm{air},2} = (1 - \Delta)/(2 + \Delta). \qquad (6)$$

### 4.2.2 Input for aerosols

The (a-priori) aerosol data for total optical depth, average single scattering albedo and asymmetry parameter (used to calculate the phase function moments) are taken from the co-located AERONET (Aerosol Robotic Network) station in Mexico City (V2, level 1.5 at http://aeronet.gsfc.nasa.gov). The a-priori shape of the profile is taken from hourly averaged ceilometer data (García-Franco et al., 2018), interpolated at the middle layer height $h$ of each layer.

The first part of MMF, the aerosol retrieval, is limited to the aerosol density profile and the total aerosol optical depth. The average single scattering albedo $\omega$ and asymmetry parameter $g$ are not subject to retrieval and are constant in all layers.

Fig. 3 (b) outlines the strategy how the aerosol contribution to the VLIDORT layer input parameters are calculated.

The hourly averages of (relative) aerosol density profile (arbitrary units) from ceilometer measurements between November 2009 to February 2013 are interpolated at measurement day time $t$ and at middle heights $h$ and provide a relative aerosol profile. This profile is scaled to match the total aerosol extinction from AERONET $\tau_{\mathrm{aer}}$. In each layer, the profile is converted into an intensive quantity by division by layer thickness, usually known as aerosol extinction profile (extinction per unit length, AE). This is then used to calculate the aerosol scattering coefficient $\sigma_{\mathrm{aer}}$ in each layer by multiplying the layer aerosol extincion with the aerosol single scattering albedo $\omega_{\mathrm{aer}}$.

The aerosol absorption coefficient $a_{\mathrm{aer}}$ is the layer aerosol extinction times $(1 - \omega_{\mathrm{aer}})$ and the aerosol phase function coefficients $\beta_{\mathrm{aer},l}$ can be calculated via the Henyey Greenstein phase function and the asymmetry parameter $g$, see Eq. 7, e.g. Hess et al. (1998), where $l$ denotes the moment,

$$\beta_{\mathrm{aer},l} = (2l + 1) \cdot g^l. \qquad (7)$$

### 4.2.3 Calculating final VLIDORT input

As mentioned in the introduction to this section, the layer input parameters for VLIDORT are total optical depth $\tau$ and single scattering albedo $\omega$ in the layer, as well as the phase function coefficients $\beta_l$. The quantities which have been calculated so far are the aerosol and air contributions to $\beta_l$ and the air and aerosol scattering coefficients $\sigma_{\mathrm{air}}$ and $\sigma_{\mathrm{aer}}$, as well as the absorption coefficients $a_{\mathrm{gas}}$ from the trace gas in question and $a_{\mathrm{aer}}$ from the aerosol in each VLIDORT input layer (red boxes in Fig. 3). These quantities need to be combined to total layer input parameters.



The total layer optical depth is simply the product of the layer thickness and the sum of all extinction and scattering coefficients:

$$\tau = \Delta h \cdot (\sigma_{\text{air}} + \sigma_{\text{aer}} + a_{\text{gas}} + a_{\text{aer}}) \tag{8}$$

The combined single scattering albedo can be calculated as

$$\omega = (\sigma_{\text{air}} + \sigma_{\text{aer}}) \cdot \Delta h / \tau \tag{9}$$

The combined expansion coefficients are calculated as follows

$$\beta_l = (\beta_{\text{air},l} \cdot \sigma_{\text{air}} + \beta_{\text{aer},l} \cdot \sigma_{\text{aer}}) / (\sigma_{\text{air}} + \sigma_{\text{aer}}) \tag{10}$$

Since Jacobians with respect to changes of quantity $\chi$ in each layer are required, the normalized derivatives of the total primary optical quantities, $\tau$, $\omega$ and $\beta$ with respect to this quantity need to be supplied. In case of trace gas retrieval, $\chi = a_{\text{gas}}\Delta h$ and in case of aerosol retrieval, $\chi = \sigma_{\text{aer}}\Delta h + a_{\text{aer}}\Delta h$. Therefore, what needs to be done is to calculate the derivatives of Eq. 8 – Eq. 10 with respect to these quantities. It should be remembered that $\beta$ is a vector.

The three quantities for aerosol are:

$$
\begin{aligned}
\frac{\chi}{\tau}\frac{d\tau}{d\chi} &= \frac{(\sigma_{\text{aer}} + a_{\text{aer}})\Delta h}{\tau} \frac{d\tau}{d(\sigma_{\text{aer}} + a_{\text{aer}})\Delta h} = \frac{\sigma_{\text{aer}} + a_{\text{aer}}}{e} \\
\frac{\chi}{\omega}\frac{d\omega}{d\chi} &= \frac{(\sigma_{\text{aer}} + a_{\text{aer}})\Delta h}{\omega} \cdot \left[ \frac{\partial\omega}{\partial\sigma_{\text{aer}}} \frac{d\sigma_{\text{aer}}}{d[(\sigma_{\text{aer}} + a_{\text{aer}})\Delta h]} + \frac{\partial\omega}{\partial a_{\text{aer}}} \frac{da_{\text{aer}}}{d[(\sigma_{\text{aer}} + a_{\text{aer}})\Delta h]} \right] \\
&= \frac{\sigma_{\text{aer}}a_{\text{gas}} - a_{\text{aer}}\sigma_{\text{air}}}{\omega e^2}
\end{aligned}
\tag{11}
$$

$$\frac{\chi}{\beta}\frac{d\beta}{d\chi} = \frac{\sigma_{\text{aer}}}{\beta} \frac{\beta_{\text{aer}} - \beta}{\sigma_{\text{aer}} + \sigma_{\text{air}}} \tag{12}$$

Here, $e$ is the total extinction coeficient: $e = \tau / \Delta h$

For the trace gas jacobian calculation, the corresponding quantities are:

$$\frac{\chi}{\tau}\frac{d\tau}{d\chi} = \frac{a_{\text{gas}}\Delta h}{\tau} \frac{d\tau}{da_{\text{gas}}} = \frac{a_{\text{gas}}}{e} \tag{13}$$

$$\frac{\chi}{\omega}\frac{d\omega}{d\chi} = \frac{a_{\text{gas}}\Delta h}{\omega} \frac{\omega}{a_{\text{gas}}} = -\frac{a_{\text{gas}}}{e} \tag{14}$$

$$\frac{\chi}{\beta}\frac{d\beta}{d\chi} = 0 \tag{15}$$

### 4.3 Calculating dSCDs and weighting functions

The forward model outputs, intensities and intensity jacobians, need to be converted into differential slant column densities and corresponding jacobians. For each set of dSCD, we have to run the forward model 2 times: Once with the gas absorption included and once without the gas absorption. Each of these sets consists of simulations in the desired telescope directions and one simulation towards the zenith. In the following, the intensity and jacobian of the simulation towards the desired angles



without gas absorption are denoted $I_0^\alpha$ and $K_0^\alpha$, the intensity (jacobian) with gas absorption towards the zenith $I_g^z$ ($K_g^z$), the intensity (jacobian) with gas absorption towards the desired angle $I_g^\alpha$ ($K_g^\alpha$) and the intensity (jacobian) without gas absorption towards the zenith $I_0^z$ ($K_0^z$). The dSCD and the corresponding weighting function $K$ can be calculated as

$$\text{dSCD} = \log\left(\frac{I_0^\alpha \cdot I_g^z}{I_g^\alpha \cdot I_0^z}\right)/(\xi_\lambda) \tag{16}$$

5   and

$$K = \frac{\left(K_0^\alpha I_g^z I_g^\alpha I_0^z + I_0^\alpha K_g^z I_g^\alpha I_0^z - I_0^\alpha I_g^z K_g^\alpha I_0^z - I_0^\alpha I_g^z I_g^\alpha K_0^z\right)}{\left(I_0^\alpha I_g^z I_g^\alpha I_0^z \xi_\lambda\right)}. \tag{17}$$

If the retrieval is to be performed in log-space, i.e. to work with $ln(x)$ instead of with $x$ as the retrieval parameter, $K$ in Eq. 17 is multiplied by $x$ and the uncertainty covariance matrix $S_a$ in Eq. 2 needs to be left and right multiplied by $1/x$. Note that the output from VLIDORT is already $Kx$, hence the immediate output from VLIDORT has to be divided by $x$ in case of linear retrieval and no extra processing step has to be performed for logarithmic retrieval.

## 5   Error analysis

The errors of gas profile retrievals in remote sensing applications can be divided into the following four different types (Rodgers, 2000):

1. Smoothing error, arising from the constraint of the $NO_2$ and aerosol profiles,

15   2. Measurement noise error, arising from the noise in the spectra,

3. Parameter error, including different parameters as aerosol profiles and cross-sections, and the

4. Forward model error, that originates from the simplification and uncertainties in the radiative transfer algorithm:

The `Gain` matrix, as defined by Eq. 18 describes the change in retrieved $x$ when measurement $y$ changes and can hence be used to map the uncertainty in measurement space, to a state-vector uncertainty.

20   $$\texttt{Gain} := \frac{\partial x}{\partial y} = (K_i^T S_m^{-1} K_i + R)^{-1} K_i^T S_m \tag{18}$$

The error analysis in general produce an error pattern $\epsilon_y$ in the measurement space of vectors of the dSCDs retrieved by QDOAS and simulated by MMF. With help of the `Gain` matrix this error pattern is then mapped as $\epsilon_x$ into the space of the solution state.





## 5.1 Smoothing error

The retrieved atmospheric state vector $x$ only represents a smoothed version of the true real state. How a change in the true atmospheric state vector $x_{true}$ is translated into changes in the retrieved state vector $x$ is described by the partial column Averaging Kernel matrix $AK_{pcol}$.

$$x - x_a = AK_{pcol}(x_{true} - x_a) + \epsilon_x \tag{19}$$

The Averaging Kernel of partial column and total column ($AK_{tot}$) describe how the retrieved solution profile depends on the real atmosphere $x_{true}$ and have to be taken into account if the profile and the column is used. The part which is not explained by $AK_{pcol}$ is explained by the retrieval error $\epsilon_x$.

As MMF uses constraint least square fitting the $AK_{pcol}$ is calculated analytically by MMF itself for each scan according to Eq.(20).

$$AK_{pcol} := \underbrace{\frac{\partial x}{\partial x_{true}}}_{\text{definition}} = \underbrace{\texttt{Gain} \cdot K}_{\text{analytical calculation}} \tag{20}$$

The Averaging Kernel matrix is in general not symmetrical and the columns are representing response functions related to a perturbation in a certain layer while the sensitivity of the $NO_2$ partial column of a certain layer (or concentration in a certain layer) to all the different true partial columns or concentrations are expressed by the rows of the matrix. If the $AK_{pcol}$ is expressed in units of partial columns (as here indicated by the subscript $_{pcol}$), the sum of the rows is the total column Averaging Kernel and represents the sensitivity of the vertical column density (VCD) to the anomalies in different heights. How the Averaging Kernel changes under transformation of the units is expressed by two matrix multiplications, one with the diagonal matrix containing the partial air column in its diagonal $U_{aircols}$ from one side and its inverse $U_{aircols}^{-1}$ from the other side.

$$AK_{VMR} = U_{aircols}^{-1} \cdot AK_{pcol} \cdot U_{aircols} \tag{21}$$

The trace of the $AK_{tot}$ matrix represents the DOF, the number of independent pieces of information in the profile retrieval, which is around 2 for the $NO_2$ retrieval.

Equation (19) separates the smoothing effect described with AK-matrices from the others errors. Before comparing the retrieved profiles of MMF with profiles from models or other measurements, such as satellite measurements, it is necessary to smooth the profile from the other source with the AK from MMF (Rodgers and Connor, 2003). Alternatively, a smoothing error $S_{smooth}$ for the profile can be calculated according to Rodgers (2000) from the a-priori covariance matrix $S_a$ and the Averaging Kernel. The latter is describing how the true atmospheric state varies, as a best estimate, while $AK = \frac{\partial x}{\partial x_{true}}$ describes how sensitive the retrieved atmospheric state vector depends on the true atmospheric state. Both quantities can be given in different





representations adjusted to the atmospheric state vector either as a fraction of the a-priori, the VMR ($_{VMR}$), as number densities or partial column profiles ($_{pcol}$) as used above.

$$S_{smooth} = (\text{AK}_{\text{VMR}} - \mathbf{1})\text{S}_{\text{a}}, \text{VMR}(\text{AK}_{\text{VMR}} - \mathbf{1})^{\mathbf{T}} \qquad (22)$$

The smoothing error for the VCD is then calculated from the full covariance matrix of the smoothing error $S_{smooth}$ using a total column operator.:

$$\sigma_{smooth} = \sqrt{g^T \cdot \underbrace{(\text{AK}_{\text{pcol}} - \mathbf{1})\text{S}_{\text{a,pcol}}(\text{AK}_{\text{pcol}} - \mathbf{1})^{\mathbf{T}}}_{\text{S}^{\text{pcol}}_{\text{smooth}}} \cdot g}, \qquad (23)$$

where **g** is the total column operator for partial column profiles: $g^T = (\,1, 1, 1, 1, 1, 1, 1, 1, 1,...1)$.

As described before, the NO$_2$ retrieval code uses a single a-priori and covariance matrix taken from the chemical transport model WRF-Chem, which results in the sensitivities and resulting smoothing errors represented in Fig. 4(a).

The aerosol retrieval, however, uses a Tihkonov constraint and no covariance matrix is available. The a-priori for the aerosol extinction profile are reconstructed by the actual total aerosol optical depth reported by the daily AERONET measurements and the average vertical distribution, reconstructed from ceilometer measurements for each hour of the day. The covariance matrix for the aerosols $Sa_{aerosol}$ is obtained by assuming a 100% variability of the used a-priori profile and an exponential correlation length of $\eta =$500 m between the different layers, as in the recent work by Wang et al. (2017).

$$Sa_{aerosol}[i,j] = (100\% \cdot \text{AE}_{\text{a}}[i] \cdot \text{AE}_{\text{a}}[j]) \exp^{(-|\text{z}[i]-\text{z}[j]|/\eta)} \qquad (24)$$

In the cases where no vertical aerosol extinction profile (AE-profile) retrieval is available, a NO$_2$ profile can still be retrieved by using an a-priori AE-profile instead of the retrieved one. Since in that case, no information of the O$_4$ absorption is used, the estimated smoothing error $S_{smooth}$ is then directly given by the a-priori covariance matrix $Sa$ for the AE-profile. The a priori information about the optical properties described by the aerosol extinction profile is designed for cloud free days and therefore the error analysis is just valid for such cloud free days.

For the calculation we have assumed a constant sensitivity AK and a constant apriori covariance matrix $Sa$. The AK of the trace gas profile indeed depends strongly on the aerosol profile and even slightly on the trace gas profile itself and the $Sa$ covariance matrix of the aerosol extinction profile should be given for each hour as the a priori profile. Therefore the estimation of the smoothing error as it is calculated here, gives just a rough general idea about the size of the smoothing error.

## 5.2 Measurement noise

The measurement noise error can be calculated from the Gain matrix and the measurement noise matrix (S$_{\text{m}}$) (Rodgers, 2000). The diagonal Matrix (S$_{\text{m}}$) is already used in the optimal estimation of the profile retrieval to weight the different dSCDs of a





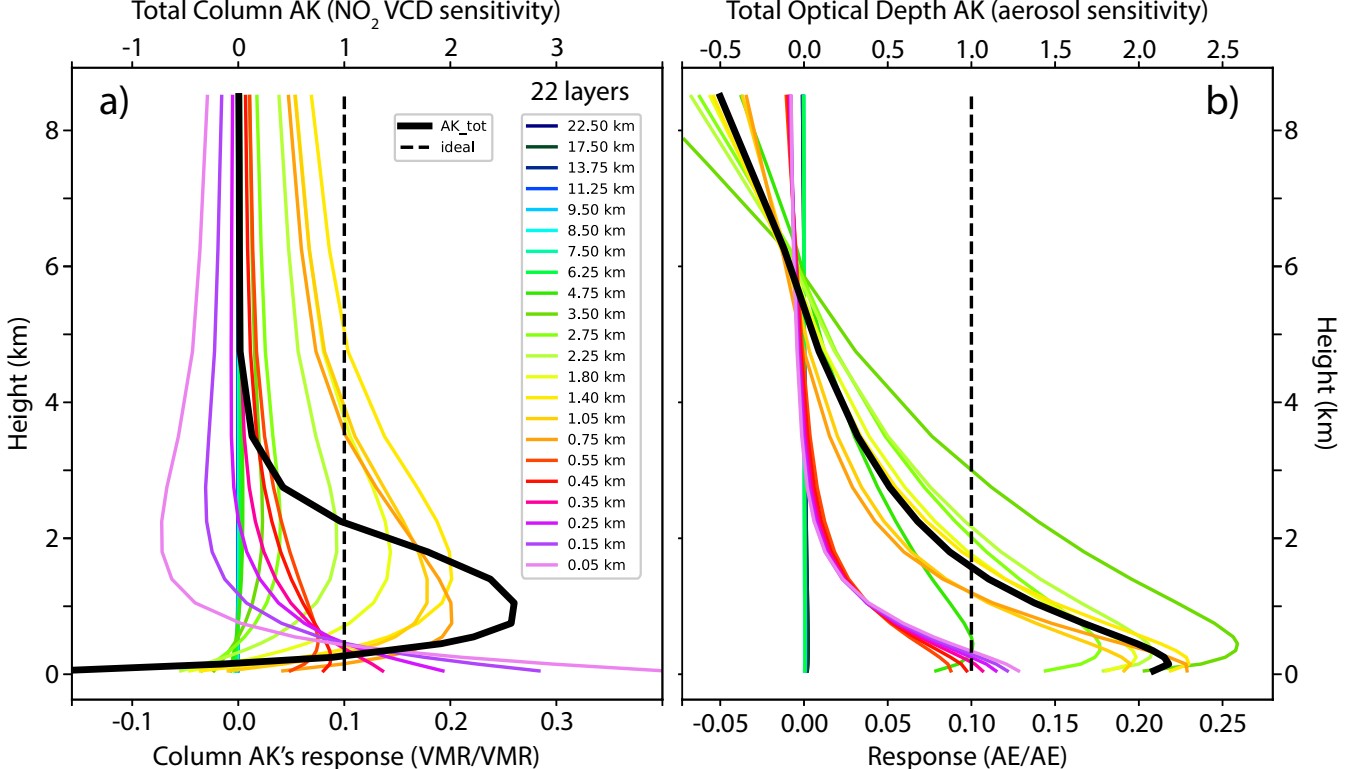

**Figure 4.** Averaging Kernel columns: (a, left) in VMR representation for $NO_2$ profile retrieval and (b,right) in the corresponding intensive quantity for the aerosol extinction (AE) profile. Each coloured line shows the expected response on perturbation in a certain layer and belongs to the lower axis. The total column Averaging Kernel: (a,left) for the $NO_2$-VCD and (b,right) for the total optical depth is given as a black line and belongs to the upper axis. The vertical dashed line in each panel indicate a ideal sensitivity of 1.0 in all altitudes.

scan according to the square of the errors from the QDOAS retrievals. The statistics of the measurement dSCDs errors and its dependence to the elevation angle has been presented in Fig. 2.

The measurement noise error matrix for both 1) the $NO_2$ profile and 2) aerosol extinction profile are calculated directly during the retrieval and are available from the MMF output of each measurement sequence as a full covariance matrix $S_{noise}$, given in units of partial columns and partial optical depths. The $NO_2$ total column error is then calculated from this and the total column operator and amounts to around 2.4% of the VCD. The errors in the profile are shown in Fig. 6.

### 5.3 Parameter errors

The parameter errors originate from all uncertainties in input-parameters in the forward model that are not properly fitted, such as the cross-sections and for the $NO_2$-profile retrieval also the used aerosol extinction profile.





### 5.3.1 Parameter error from Spectroscopy

The error originating from the cross-section is estimated by assuming that the column amount regarding to the used cross-section has a uncertainty of 3% (Wang et al., 2017). Therefore, the error can be calculated using generalized measurement error covariance matrix $S^y_{spectroscopic}$ in the measurement space assuming that all retrieved dSCD are 3% too high (or low)

and using the outer product of the measurement state vector containing the retrieved dSCDs.

$$S^y_{spectroscopic} = (0.3\%)^2 \cdot yy^T \tag{25}$$

$$S^x_{spectroscopic} = \mathrm{Gain} \cdot \mathrm{S^y_{spectroscopic}} \cdot \mathrm{Gain}^T \tag{26}$$

From the $S^x_{spectroscopic}$ matrix the error in the total column and profile are calculated as shown earlier in Sec.5.1. As expected the error for the total column is 3% and the errors in the $NO_2$ VMR-profile are shown in Fig. 6. The spectroscopic error is a

purely systematic error and affects all retrieval in the same manner.

### 5.3.2 Parameter error from aerosol profile

The AE-profile is crucial for the $NO_2$-profile retrieval because of its strong contribution to the airmass factor. The uncertainties in this vector of input parameters arise from errors in the aerosol extinction profile retrieved from the spectral signature of the $O_4$-dimer. For the propagation of the error of the $O_4$-AE-retrieval into the $NO_2$-retrieval we assume here that the total error in

the AE-profile depends on just the two contributions a) smoothing error and b) measurement noise errors.

The effect that the assumed aerosol content in each layer has on the retrieved $NO_2$ profile is calculated by a sensitivity study. For this, 22 $NO_2$ retrievals are performed for the same measurement sequence (dSCDs) assuming slightly modified aerosol extinction profiles. First, a normal retrieval with the best estimated aerosol extinction profile is retrieved. Then, the aerosol extinction profile is modified so that the partial extinction of the i-th layer is increased by 1% of the total optical depth with

respect to the original aerosol extinction profile.

The difference between the perturbed and original $NO_2$ VMR profiles are combined into the matrix D, describing thus how the $NO_2$ VMR profile responds to changes in the aerosol at different heights. The result is presented in Fig. 5.

With the help of the matrix $D_{VMR}$, the different errors in the aerosol extinction profile as measurement noise, smoothing or even due to the algorithm error according to Wang et al. (2017) can be propagated to the $NO_2$ profile and its VCD.

$$S^{aerosol}_{NO2-VMR} = D_{VMR} Sa_{aerosol} D_{VMR}{}^T \tag{27}$$

$$error^{aerosol}_{NO2-VCD} = \sqrt{g^T U_{aircols} \cdot D_{VMR} \cdot Sa_{aerosol} \cdot D_{VMR}{}^T U_{aircols} \cdot g}, \tag{28}$$

The propagation of the smoothing (4.6%) and measurement noise (2.2%) errors of the $O_4$ retrieval into the $NO_2$-retrieval results in a 5.1% error in the $NO_2$ VCD, while if no $O_4$-retrieval is performed successfully the error would be in our example





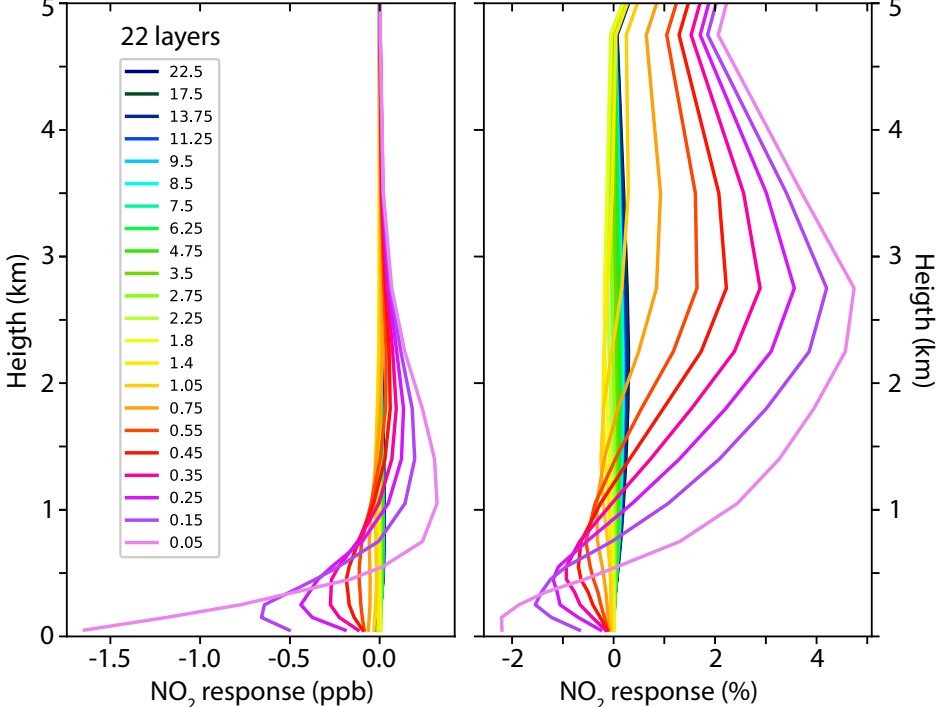

**Figure 5.** Plots showing the results of how sensitive the $NO_2$ profile is to changes in the aerosol extinction profile. Left: columns of the matrix D created to describe the response in the $NO_2$ VMR profile (the differences with respect to the original retrieval). Right: same as left but the response to the perturbation in the aerosol profile is given as fraction of the retrieved $NO_2$ profile.

9.8 %. In case we would include the algorithm error (7.8%) introduced by Wang et al. (2017) the error when a $O_4$-retrieval is performed successfully would be 9.4%. However the algorithm-error is calculated from the resulting residual of the fit and is not independent on the other error sources as mentioned earlier.

### 5.4 Forward model error

5  The forward model error could be evaluated if an improved forward model was available, which it is not the case. However errors in the forward model would result in systematic structures in the residual and a larger residual than expected from the error calculated by QDOAS for the slant columns. The most rigorous simplification in the forward model is the assumption of a horizontal homogeneity of gas and aerosols. A horizontal inhomogeneity leads to a set of slant columns in a scan which cannot be simulated by the VLIDORT and the residuals of measured minus calculated slant columns indicate this error. Following
10  Wang et al. (2017) we calculate the variance in the residuals as a function of viewing angle and hence the algorithm error. But as all errors increase the residual the so called *algorithm error* is not identical to the forward model error. However, it is maybe a good way to check empirically if there are important error contributions missing in the analytical analysis.



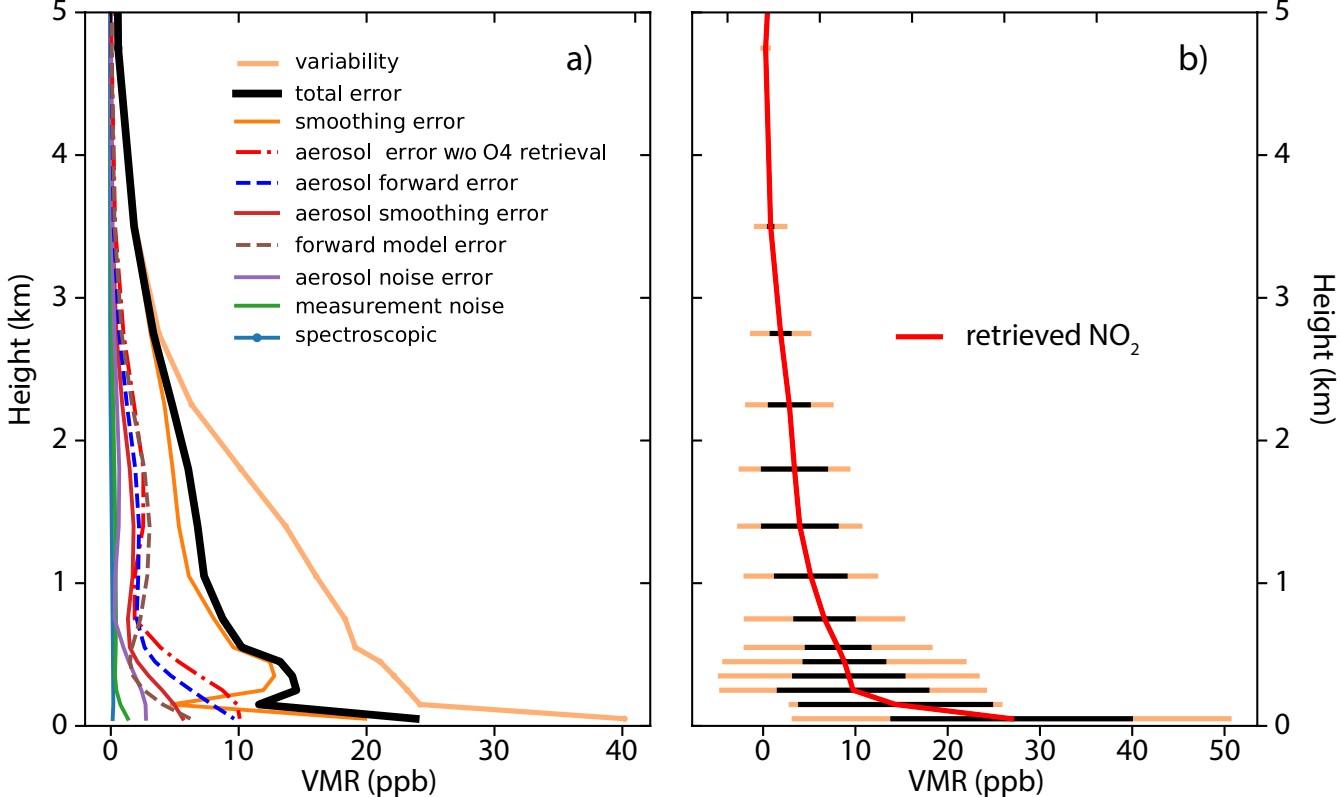

**Figure 6.** Altitude-dependent errors in a $NO_2$ VMR profile measured at 13:58 on May 20, 2016 (UTC-6h). a) square root of the diagonal in the covariance matrix describing the individual error contributions in the VMR profile and b) the retrieved $NO_2$ VMR profile with corresponding error bars (total. error and variability).

## 5.5 Total error

The results from the error estimations are summarized in Table. 1. The overall error in the $NO_2$ VCD is estimated to be around 14.1% (20.3% including the algoritm errors ($NO_2$ and $O_4$) calculated from the residuals). The contributions are 12.5% from smoothing, 2.4% from measurement noise, 3% from spectroscopy and 5.1 % from errors in the aerosol extinction profile. The 
5  algorithmn error are 12.3% from the $NO_2$ retrieval itself and 7.8% from the $O_4$-AE retrieval. These results are fairly similar to those reported by Wang et al. (2017).

    The error in the vertical column is relatively smaller than the errors in the VMR profile for almost all layers (Fig.6) as there is an anti-correlation in different partial column errors indicated by to the full error covariance matrix. While the measurement noise seems to play a minor role, the smoothing and the aerosol profile are the main sources of error. Even though the error 
10  might seem large, the retrievals still provide new and relevant information of $NO_2$ within the boundary layer of Mexico City.



| | Error | Description | $NO_2, \%$ |
|---|---|---|---|
| 1 | Smoothing | Variability from model | 12.6 |
| 2 | Measurement noise | Fig. 2 | 2.4 |
| 3 | Spectroscopy | 3 % | 3 |
| 4 | from $NO_2$ residuals | 20 residuals (diagonal) | (12.3) |
| 5 | Aerosol noise | from QDOAS | 2.2 |
| 6 | Aerosol smoothing | Wang et al.,(2017) | 4.6 |
| 7 | Aerosol variability | Wang et al.,(2017) | (9.8) |
| 8 | from $O_4$ residuals | 20 residuals (diagonal) | (7.8) |
| 9 | Total aerosol | 5,6 | 5.1 |
| 10 | Total aerosol | 5,6,8 | 9.3 |
| 11 | Total, including | 1,2,3,5,6 | **14.1** |
| 12 | Total, including | 1,2,3,7 (without $O_4$ retrieval) | 16.4 |
| 13 | Total error (all) | 1,2,3,4,5,6,8 | 20.3 |

**Table 1.** Errors in $NO_2$ vertical column density as fraction of a retrieved VCD of 3.2e16 molec./cm² measured at 13:58 on May 20 of 2016 (UTC-6h). The total error is calculated as the square root of the sum of the squares of different independent errors. Different total errors are calculated by including or not including the algorithm error (Wang et al., 2017) and for the two cases that an aerosol extinction profile is retrieved from $O_4$-absorption or just an a-priori guess is used. For the assumed variability of the aerosol extinction the error due to the uncertainty remaining after an aerosol extinction profile retrieval is 5.6% instead of 9.8%. But if the algorithm error according to Wang et al. (2017) is included the remaining error due to the uncertainty in the aerosol profile is slightly better 9.3% instead of 9.8% without $O_4$ retrieval.

# 6 Results

In this section we present results of the $NO_2$ variability measured in one of the stations comprising the MAX-DOAS network operated in Mexico City, and compare them with *in situ* measurements performed at the surface. A data set from January 2015 to July 2016 (aprox 19 months) is considered in this study, in which both MAX-DOAS and *in situ* surface measurements were available. A total number of 2531 coincidences of hourly averages were found in this period, considering that at least 6 measurement sequences of the MAX-DOAS instrument in each reported hour needed to be available. The complete time series is presented in Fig. 7, showing hourly $NO_2$ averages of the *in situ* surface concentration in red, the total vertical column from the MAX-DOAS in blue, and the average VMR of its first 6 layers closest to the surface in green. This plot includes all MAX-DOAS results regardless of the sky conditions.

Apart from the large data gaps in the beginning of 2016, the time series is quite complete and the fitted annual periodic functions (a Fourier series constrained to a fixed seasonal cycle shape) applied to the three data sets show a similar pattern. VMR values from the surface measurements are clearly higher than the VMR's detected in the lowest layers of the MAX-DOAS retrievals.





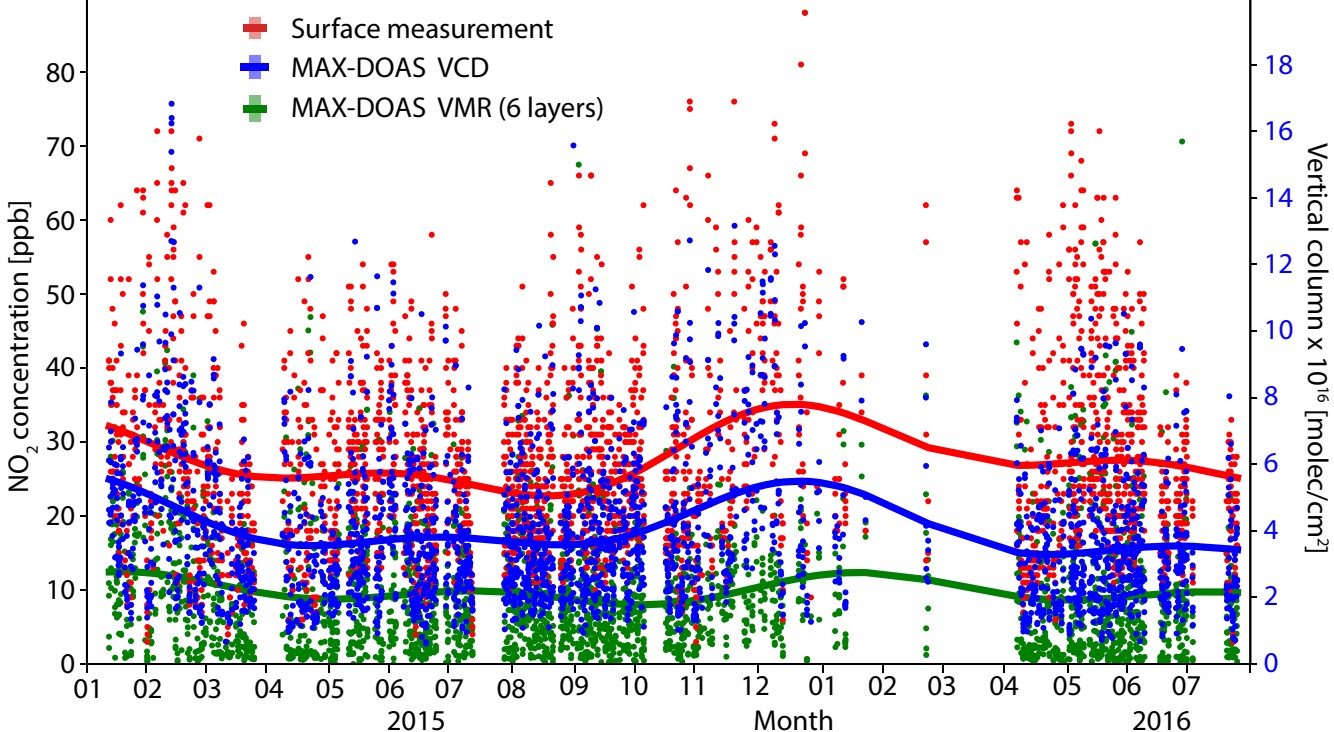

**Figure 7.** Hourly $NO_2$ averages of *in situ* surface concentrations (red) in ppb, total VCD's from the MAX-DOAS instrument (blue) in molecules/cm$^2$ and the average VMR of the first 6 layers (up to 550 m) in the retrieved MAX-DOAS profiles (green), also in ppb.

The 2531 coincidences were correlated averaging different number of layers starting from the ground and the resulting Pearson's coefficients (*R*) are plotted in Fig. 8 (top). Currently, the integration times in the spectra from which the $O_4$ dSCDs are calculated, are not long enough to ensure an $O_4$ dSCD error resulting in DOF larger than 1 for the aerosol retrieval. Since we use a Tihkonov regularization for aerosol retrieval, this means that we can basically retrieve the total aerosol extinction. Due to

5 the topographical circumstances in Mexico City, where the boundary layer rises steadily during the day, using an a-priori that is calculated from hourly averages of ceilometer data will generally provide a good a-priori profile shape. However, for days that likely have a very different aerosol profile from our a-priori profile, the aerosol profile included in the trace gas forward model accounts for far more than the estimated error in 5.3.2 since this estimation uses a fixed percentage of the a-priori profile in the $S_a$ matrix. For cloudy days, for example, the form of the profile will look considerably different than the ceilometer averages.

10 We are currently not able to retrieve such profiles. Hence, we limit our analysis to cloud free days. Currelty, the retrieval chain has no cloud-screening algorithm included. As an ad-hoc solution, we use the presence of AERONET data on that day as an estimator for cloud free days. For our entire data set, about in half of the coincident measurements (1270) there was at least one AERONET measurement available on that particular day while for the rest of our data (1261), no AERONET data was available on that day.





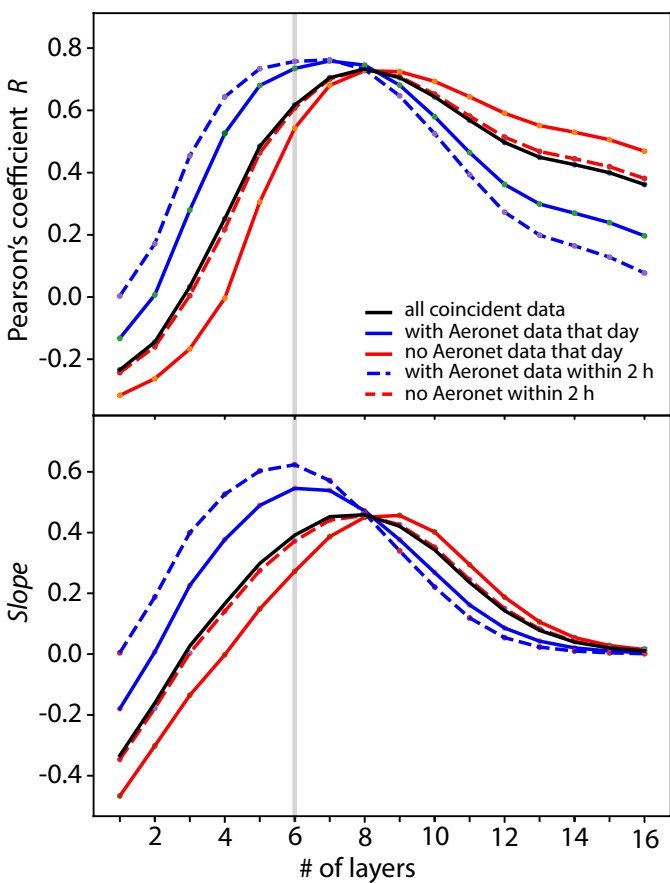

**Figure 8.** Pearson's correlation coefficients (top) obtained from hourly surface and MAX-DOAS coincident measurements as a function of the number of layers considered in the VMR averages. The slopes obtained from the linear regressions, performed forcing the intercept to be at zero to the same data sets, are shown in the lower panel. The layers below layer 6 are very likely inside the mixed boundary layer, while layers above layer 6 might be on the edge or above the boundary layer during some hours in the morning (Franco-Garcia et al., 2018). The blue dashed lines only show data when AERONET data are avaible within 2 hours of the scan and hence limit the measurements to cloud free conditions. This selection criteria improves the correlation significantly.





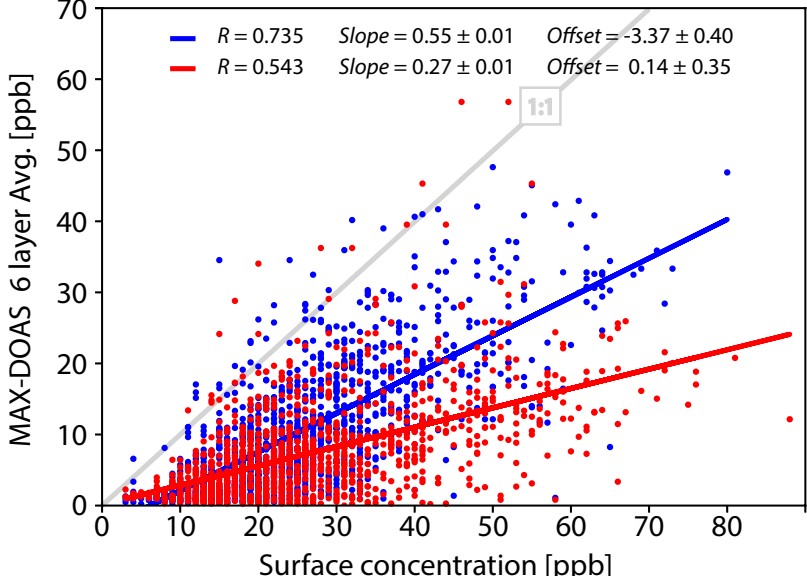

**Figure 9.** Correlation plot of the MAX-DOAS 6-layer VMR averages vs. the surface measurements shown separately for clear sky measurements in which aerosol data on that day was available (blue) and for MAX-DOAS retrievals with no AOD availability from AERONET, corresponding most likely to cloudy days (red).

Generally, a good correlation is found when averaged MAX-DOAS VMR's are compared to *in situ* surface measurements. Average $R$ values larger than 0.7 are obtained in all cases when the first 6-8 layers are considered in the correlation. These correlations decrease rapidly when less layers are considered due to increased errors in the lowermost part of the profile. Naturally, the $R$ value also decreases towards larger number of layers averaged since different air masses are measured at

5    higher altitudes. As expected, slightly larger correlations are obtained for MAX-DOAS retrievals using a measured aerosol input from the AERONET (blue traces in Fig. 8), which correspond to clear sky conditions.

Linear regressions as the ones shown in Fig. 9 were generated for each data set, in order to gain more information as to how the number of layers averaged relates to the VMR comparison with the *in situ* surface measurements. The slope of all coincident data using the first 7-8 layers is 0.45, and it can reach values above 0.6 considering the first 6 layers in clear sky

10    conditions (AERONET data available within ±1 h of the measurement time). We thus decide to use in this study the 6-layer VMR averages of our MAX-DOAS retrievals, reaching a height of 550 m above the ground level, in the comparisons we present with surface concentrations.

To investigate the difference in our data set between clear and cloudy days, a correlation plot is presented in Fig. 9 showing the linear regressions produced from the MAX-DOAS 6-layer VMR averages for the retrievals with AERONET availability

15    on that day in blue and those retrievals using a default aerosol a-priori in red. A significant improvement in the correlation coefficient going from 0.54 to 0.74 is evident in the plot. The $R$ and slope values when all the coincident data is considered, indistinctly if the retrieval had data available from the AERONET instrument on that day or not, are 0.62 and 0.39, respectively.



Examples of how the averaged 6-layer VMR's and VCD compare to the ground level concentrations in individual days are presented in Fig. 10. These examples were chosen to depict distinct diurnal patterns that occur in different times of the year. Again, the MAX-DOAS 6-layer product is consistently lower than the $NO_2$ concentrations measured at the surface. These, together with the corresponding VCD's plotted on a different y-axis, follow the pattern of the surface measurements quite well. The MAX-DOAS instrument captures the features observed at the surface but not without some interesting differences.

May 4, 2016 was in the middle of a high pollution episode in Mexico City, in which a restriction on the use of private vehicles was declared during four days. Ozone had surpassed, at least in one of the stations from the monitoring network operated by the city government, the 165 ppb 1-hour average concentration. It is interesting to see in the May 4 plot, that indeed the surface concentrations of nitrogen dioxide were high in the morning, a condition favoring ozone production, but there is a large $NO_2$ peak just after noon in the MAX-DOAS VCD and 6-layer product data which is not detected at the surface. This is more evident from the individual measurements shown as dots than from the hour averages.

Other differences have to do with the evolution of the mixed layer height, which has been shown to have a rapid growth in the late morning, strongly influencing the surface concentrations (García-Franco et al., 2018). This is for example evident on the December 22 plot, where a peak is observed at 11 h and a second one at 14 h. This peak is strongest at 11 h in the case of the surface measurement, but the total column shows the peak at 14 h to be dominant. The mixed layer has grown in the mean time so that the registered 14 h surface concentration is relatively lower despite that there are more $NO_2$ molecules in the atmosphere.

In order to see the diurnal and seasonal patterns from both MAX-DOAS and surface measurements, all coincident measurements between January 2015 to July 2016 including days with clouds were averaged to a specific hour or month, respectively. For consistency, only the coincident 1-hour data was considered in order to make the data sets comparable. The mean diurnal evolution with standard deviations as vertical bars are shown in Fig. 11. The maxima of both surface and MAX-DOAS 6-layer products are at around 10 h, while that of the vertical column is shifted towards noon. This is expected from what is known from the emissions, growing mixed layer and ventilation patterns in Mexico City.

Fig. 12, on the other hand, presents the compiled data considered in this study as monthly means in order to observe the seasonal variability and compare the data sets from these two measurement techniques. The three data products coincide in having the highest monthly mean values between April and June. A difference which is probably interesting to note is during December and January, when the concentrations from the surface measurements are relatively higher than those from the MAX-DOAS products. This might have to do that during these winter months the mixed layer is shallower than during the rest of the year, and the sensitivity of the MAX-DOAS instrument is reduced for the lowest layers as described above.

# 7 Conclusions

In this contribution we describe the methodology used to analyze the data produced by the network of MAX-DOAS instruments (Arellano et al., 2016) operating in the Mexico City Metropolitan Area. In general terms, MMF is a retrieval code developed to process the data acquired by the instruments converting the measured spectra together with other input parameters to vertical




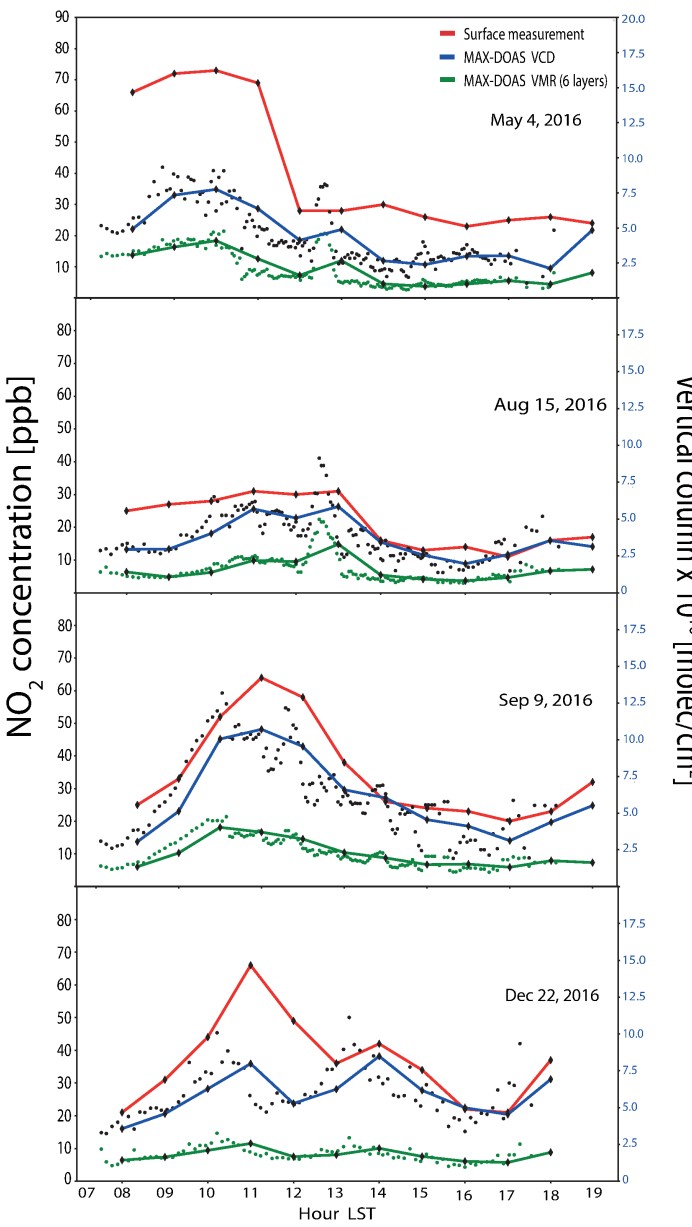

**Figure 10.** Example days showing the NO₂ variability in both VMR from the MAX-DOAS (green, < 550 m) and surface measurements (red), as well as the total vertical column densities (blue).





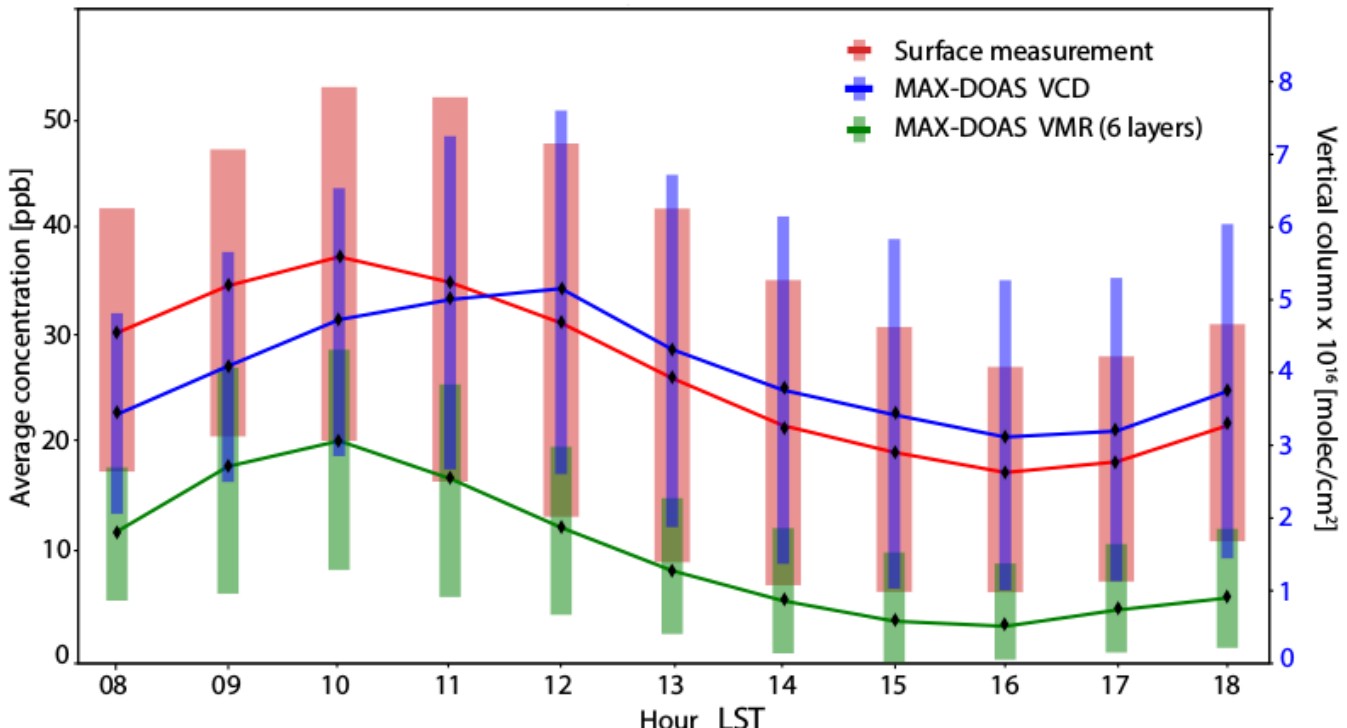

**Figure 11.** Diurnal $NO_2$ variability from hourly means of the entire data set spanning from January 2015 to July 2016. Only the data with coincident surface and MAX-DOAS measurements are included for consistency in the comparison. Vertical bars correspond to the standard deviations.

profiles of aerosols and target gases. We have tested the performance of the code for $NO_2$ in one of the stations and present some diagnostics and a full error analysis of the results. In the case of $NO_2$, all error sources amount to about 14.1%, the smoothing error being the dominating one (12.5%) followed by the error due to the uncertainties in the aerosol extinction profile (5.1%).

5      Both the resulting total vertical column densities and a product consisting of the average VMR in the first 6 layers (< 550 m above the ground level), were compared to ground level $NO_2$ concentrations. Good correlations were obtained between the 6-layer product and the values from the surface measurements, with *R* values between 0.6 and 0.7. However, the MAX-DOAS systematically underestimates the ground level concentrations by a factor of about 0.4. This is consistent with the total column averaging kernels reported in section 5.1, meaning that the MAX-DOAS instrument has a significantly lower sensitivity near

10    the surface and is most sensitive at a height of around 1 km. It is shown, however, that this underestimation is less for clear sky conditions as suggested from comparing data when aerosol optical depths where available from independent measurements.

     Although results are shown only from one instrument located in the southern part of the city (UNAM), several years of data are available from three other stations at different locations within the metropolitan area (Acatlán, Vallejo and Cuautitlán)



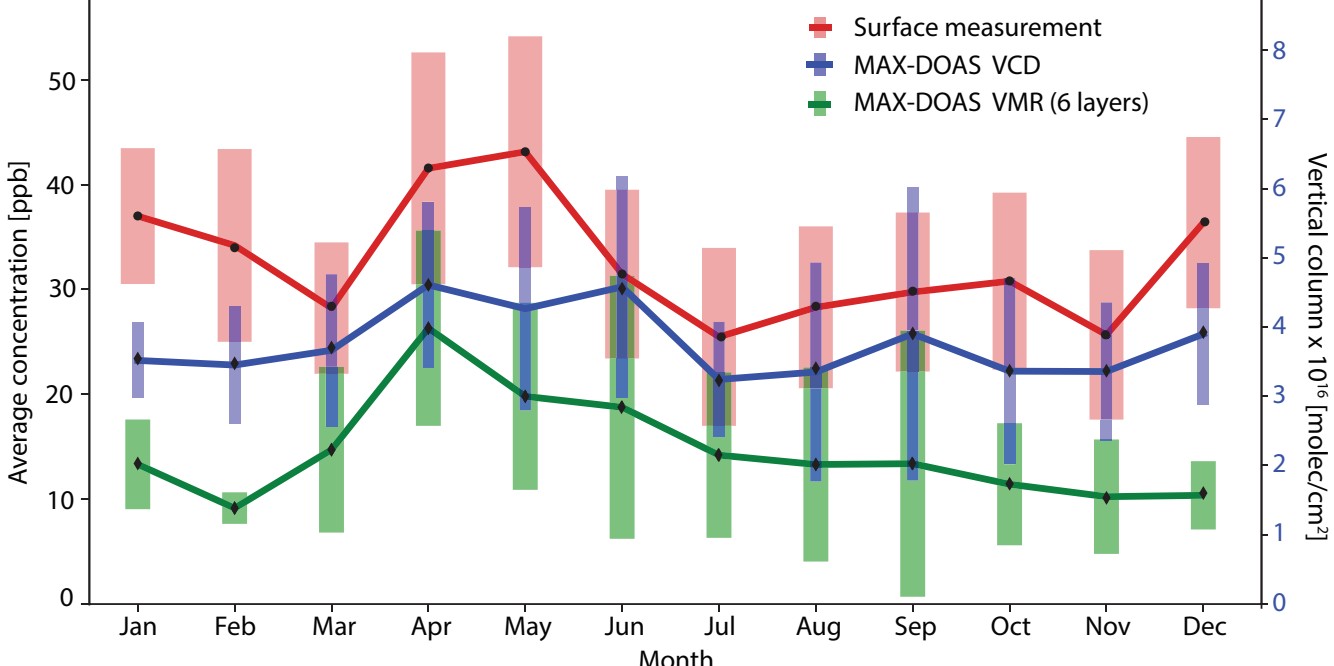

**Figure 12.** Annual $NO_2$ variability resulting from the monthly mean coincident data (surface and MAX-DOAS) between January 2015 and July 2016. The standard deviations are given in vertical bars.

that are being analyzed and used for studying the spatial and temporal variability of $NO_2$'s in conjunction of several satellite products.

In this work we present a new and competitive retrieval code which has been developed for retrieving trace gas profiles from MAX-DOAS measurements. It has been further improved to perform retrievals in logarithmic space to avoid unphysical

5  negative partial columns and oscillations; it uses the more stable Levenberg-Marquardt iteration scheme and decouples the retrieval and simulation grids.

Further efforts include the improvement in the measurement noise by increasing the integration times during spectral acquisition, which will allow us to improve the retrieval of other gases such as formaldehyde (HCHO) and other weak absorbers. This will also enable us to use OE as retrieval strategy for aerosol retrieval and possibly increase the DOF to comparable

10  values as for $NO_2$ retrieval. This, together with an inclusion of a cloud-screening algorithm in the retrieval chain, will make the retrieval less dependend on the availability of AERONET data.

## Appendix A: Calculations for dry air density

In this appendix, we outline the calculation of $Z$, the molar mass of air $M_a$ and the depolarization ratio $\Delta$. We follow Ciddor (1996). If $T$ is given in K (temperature in C is denoted as $T_C$), $\sigma$ the wavenumber in $1/\mu$m, pressure $P$ in Pa, volume mixing



ratio of $CO_2$ in ppm, the constants in Eq. A1 - Eq. A14 have the following values:

| | | | |
|---|---|---|---|
| $k_0 = 238.0185$ | $a_0 = 1.58123 \times 10^{-6}$ | $b_0 = 5.707 \times 10^{-6}$ | $w_0 = 295.235$ |
| $k_1 = 5792105.0$ | $a_1 = -2.9331 \times 10^{-8}$ | $b_1 = -2.051 \times 10^{-8}$ | $w_1 = 2.6422$ |
| $k_2 = 57.362$ | $a_2 = 1.1043 \times 10^{-10}$ | $c_0 = 1.9898 \times 10^{-4}$ | $w_2 = -0.032380$ |
| $k_3 = 167917.0$ | $\alpha = 1.00062$ | $c_1 = -2.376 \times 10^{-6}$ | $w_3 = 0.004028$ |
| $d = 1.83 \times 10^{-11}$ | $\beta = 3.14 \times 10^{-8}$ | $A = 1.2378847 \times 10^{-5}$ | $C = 33.93711047$ |
| $e = -0.765 \times 10^{-8}$ | $\gamma = 5.6 \times 10^{-7}$ | $B = -1.9121316 \times 10^{-2}$ | $D = -6.3431645 \times 10^{3}$ |

First, the saturation vapor pressure $S$ is calculated according to Eq. A1 then the enhancement factor for water vapour in air, $\mathcal{F}$, is calculated according to Eq. A2 and with this, the molar fraction of water vapor in moist air $x_w$ is calculated according to Eq.A3 where $P_w$ is partial water vapor pressure in Pa.

$$S = \exp\left(AT^2 + BT + C + D/T\right) \tag{A1}$$

$$\mathcal{F} = \alpha + \beta P + \gamma T_C^2 \tag{A2}$$

$$x_w = \mathcal{F} P_w / P \tag{A3}$$

The air density in dry air $n_{\text{air,dry,450}}$ (0 % humidity) at standard conditions (15 °C, 101.325 Pa, $xCO_2$ = 450 ppm ) can be calculated as in Eq. A4 and from this, the dry air density at $xCO_2$ is calculated according to Eq. A5.

$$\left(n_{\text{air,dry,450}} - 1\right) \times 10^8 = \frac{k_1}{k_0 - \sigma^2} + \frac{k_3}{k_2 - \sigma^2} \tag{A4}$$

$$\left(n_{\text{air,dry,}xCO_2} - 1\right) = \left(n_{\text{air,dry,450}} - 1\right) \times 10^8 \left(1.0 + 0.534 \times 10^{-6} \cdot (xCO_2 - 450.0)\right) \tag{A5}$$

The equivalent quantity for water vapour, $n_{\text{wv}}$, is calculated as follows:

$$n_{\text{wv}} = 1.022 \cdot \left(w_0 + w_1 \sigma^2 + w_2 \sigma^4 + w_3 \sigma^6\right) \tag{A6}$$

The molar mass of dry air, $M_a$ with $xCO_2$ is calculated as:

$$M_a = 10^{-3} \left(28.9635 + 12.011 \times 10^{-6}(xCO_2 - 400.0)\right) \tag{A7}$$

The compressibility of dry air, $Z_a$, and pure water vapour, $Z_w$, are calculated as:

$$Z_a = 1.0 - \frac{101325.0}{288.15}\left(a_0 + 15 a_1 + 225 a_2\right) \tag{A8}$$

$$Z_w = 1.0 - \frac{1333.0}{293.15}\left(a_0 + 20 a_1 + 400 a_2 + b_0 + 20 b_1 + c_0 + 20 c_1\right) + \left(\frac{1333.0}{293.15}\right)^2 (d + e) \tag{A9}$$

Using Eq. A10 once with the values for $Z$, $R$ and $T$ for standard water vapour and once with the values for standard air, one can calculate $\rho_{\text{ws}}$ and $\rho_{\text{axs}}$, respectively.

$$\rho = (p M_a/(Z R_{gas} T))\left(1 - x_w(1 - M_{\text{mwv}}/M_a)\right) \tag{A10}$$





**Table A1.** depolarization ratios and VMR for main air constituents

| gas | VMR/ppm | $\Delta$ gas |
|-----|---------|--------------|
| $N_2$ | 780840.0 | see Eq. A16 |
| $O_2$ | 209460.0 | see Eq. A16 |
| Ar | 9340.0 | 0.0 |
| $CO_2$ | 400.0 | 0.0814 |

Here, $R_{gas} = 8.3144621$ J/mol/K is the gas constant, $M_{mwv} = 0.018015$ kg/mol is the molar mass of water vapour. $Z$, the compressibility of moist air at $T$, $x_w$ and $P$ can be calculated as

$$Z = 1.0 - (P/T)(a_0 + a_1 T_C + a_2 T_C^2 + (b_0 + b_1 T_C)x_w + (c_0 + c_1 T_C)x_w^2) + (P/T)^2 (d + e\, x_w^2) \tag{A11}$$

Using again Eq. A10, with the actual values for $P$, $x_w$ and $Z$, the density of the dry air component can be calculated according to Eq. A12 and the density of the water vapour component can be calculated according to Eq. A13

$$\rho_{\text{dry air}} = P M_a (1 - x_w)/(Z R_{gas} T) \tag{A12}$$

$$\rho_{\text{wv}} = P M_{mwv} x_w/(Z R_{gas} T) \tag{A13}$$

Finally, the number density of the air $\rho$ and the refractive index $n$ are calculated as:

$$\rho = \frac{\rho_{\text{dry air}}}{M_a} A_v \tag{A14}$$

$$n = \frac{\rho_{\text{dry air}}}{\rho_{\text{axs}}} (n_{\text{air,dry,xCO}_2} - 1) + \frac{\rho_{\text{wv}}}{\rho_{\text{ws}}} (n_{\text{wv}} - 1) \tag{A15}$$

Here, $A_v = 6.02214129 \times 10^{23}$ is the Avogadro number.

We use the $N_2$, $O_2$ and $CO_2$ depolarization factors from Bates (1984) and fixed VMR as summarized in Table A1 and calculate the wavelength dependent depolarization factor via Eq. A16 where $F$ is the so called King factor and is calculated for $N_2$ ($\lambda$ in $\mu$m) and $O_2$ according to Eq. A18 and Eq. A19, respectively. The complete depolarization factor is then calculated according to Eq. A19

$$\Delta = 6.0 \cdot (1.0 - F)/(-7.0 F - 3.0) \tag{A16}$$

$$F(N_2) = 1.034 + 3.17 \times 10^{-4}/\lambda^2 \tag{A17}$$

$$F(O_2) = 1.096 + 1.385 \times 10^{-3}/\lambda^2 + 1.448 \times 10^{-4}/\lambda^4 \tag{A18}$$

$$\Delta = \sum (\Delta_i \text{VMR}_i), \quad i = N_2, O_2, Ar, CO_2 \tag{A19}$$





For the Rayleigh scattering cross section, we implement equation 7.37 from Goody and Yung (1989) and equation 19.5 from Platt et al. (2007), see Eq. A20.

$$Q_{\mathrm{Ray}} = \frac{24.0\pi^3 (n^2 - 1.0)^2}{\lambda^4 \, \rho^2 \, (n^2 + 2)^2} \frac{6.0 + 3.0\,\Delta}{6.0 - 7.0\,\Delta} \tag{A20}$$

*Author contributions.* MMF: part of Abstract and Section 1, Figure 1, Figure 2, Figure 3, Section 4, Appendix, retrieval code development and testing, retrieval chain setup from spectra to profiles, retrieval parameter choice for MMF, software support CR: parts of Section 2, Section 3. QDOAS setup, dSCD retrieval parameter choice, technical support WS and ZO: parts of Abstract, Section 5, Figures 4 – 12, part of Sect. 6 and 7. data retrieval, data analysis and interpretation, sensitivity tests and complete error analysis. JA: Section 2, instrument construction, installation and maintenance AB: parts of Sect. 2, technical support and data management support, instrument maintenance JAGR: part of Sect. 4.2.1, WRF-Chem simulations that serve as a-priori data for $NO_2$ retrieval MG: Sect. 1, Sect 6 and Sect. 7., data analysis and interpretation.

*Competing interests.* The authors declare that they have no conflict of interest

*Acknowledgements.* We would like to thank the financial support provided by CONACYT (grants 275239 & 290589) and DGAPA-UNAM (grants IN107417,IN111418 and IA100716 & M.M. Friedrich's postdoctoral fellowship). SEDEMA is acknowledged for providing the *in situ* surface concentrations from their monitoring network (RAMA) as well for partially supporting the MAX-DOAS network, and A. Krueger, A. Rodríguez, H. Soto and D. Flores are thanked for their technical assistance. We thank Thomas Danckaert (thomas.danckaert@aeronomie.be), C. Fayt (caroline.fayt@aeronomie.be) and M. Van Roozendael (michelv@aeronomie.be) for the free use of the QDOAS software and we thank R. Spurr for free use of the VLIDORT radiative transfer code package.



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
