# Peer review of "NO2 vertical profiles and column densities from MAX-DOAS measurements in Mexico City"

_Atmospheric Measurement Techniques, 2018_

## Referee Comment (RC1) · Anonymous Referee #2 · 9 Jan 2019

**Comment on NO$_2$ vertical profiles and column densities from MAX-DOAS measurements in Mexico City**

The authors describe a new MAX-DOAS profile retrieval code, the Mexican Maxdoas Fit (MMF), including an examination of the measurements, the algorithmic approach, and the error budget. The code is then applied to an 18 month dataset from Mexico City and compared to an in situ sensor. When averaged over the boundary layer MAX-DOAS measurements are found to capture diurnal and seasonal trends, but are systematically lower than the in situ measurements. The patterns in the mismatch can be understood to in part reflect meteorology.

At its most basic, the manuscript does two separate but related things: it describes MMF and it examines the 18 months of data in Mexico City using MMF. The fact that MMF has been updated since it was applied to the 18 month data set presents something of a challenge in presenting both the code and the data set clearly. Some restructuring of the content would better provide transparency and clarity to the reader. Further, some minor additions to the figures and text could make the message of the manuscript more coherent and relevant. I offer specific comments below.

Major comments:

**1)** Regarding the dSCD retrieval and Section 3:

**a)** The authors use a zenith measurement prior to the scan to analyze the scan. If the upper atmospheric contribution to the dSCDs changes during the scan this can lead to signal in the measurements which is from the upper atmosphere being falsely attributed to lower altitudes especially in low eastward elevation angles at the end of the scan. The effect would be expected to lead to lower VCDs in the morning and higher VCDs in the evening, especially in winter. Because of the short 7 minute scan time this effect would likely only be significant at twilight.

Does the instrument acquire data at twilight which are included in the analysis? Can the authors bound the impact of such an effect and compare it to the magnitude of their error budget?

**b)** For the fitting setting in the retrieval the authors use older cross-sections where newer cross-sections for the same gases are increasingly standard in the community e.g. (Damadeo et al., 2013; Peters et al., 2017). For O$_3$ they use (Burrows et al., 1999) rather than (Bogumil et al., 2003) or (Serdyuchenko et al., 2014) and for O$_4$ they use (Hermans et al., 1999) rather than (Thalman & Volkamer, 2013). Was there any particular reason for these choices? Is it based on Orphal, 2002 cited later?

**2)** Regarding the profile retrieval method description and Section 4:

**a)** At present the aerosol and trace gas inversions in Fig. 1 are presented as the same, whereas the former uses Tikhonov regularization and the latter optimal estimation. This should be reflected in the figure as it is in the text.

**b)** The code used to analyze the 18 month data set presented in the work utilized a Gauss-Newton (GN) iteration scheme for inversion, however, MMF has since been updated to utilize a Levenberg Marquardt (LM) iteration scheme, as well as other more minor updates. At present both schemes are described somewhat in parallel, and the authors are diligent in describing which scheme they are discussing. Nonetheless, equations for the GN scheme, which was used, are sometimes left out in favor of the more current LM scheme equivalents, leaving the methods applied not fully transparent to the reader.

I would recommend describing the GN scheme as default as it is most relevant to the titular topic of the work and collecting and describing the changes for the LM scheme either all together in a dedicated section or within the relevant subsections.

**c)** At the top of page 8 is the following paragraph :

"For the aerosol retrieval used in this study, we use the L1 operator (R = L1$^T$ α L1) where the scaling parameter α is supplied via an input script to limit the degrees of freedom (DOF) to just slightly above 1. Different scalings for the upper layers and lower layers can be supplied, as well as a complete regularization matrix R."

I understand the latter sentence to describe a capability of MMF, but how was the regularization conducted for the analysis presented later? Was a constant α determined such that the DOF was just over 1 or was something else done?

**d)** Discussing the advantages of MC RTM codes, the ability to model statistically rare photons and output information of the distribution of photons is also useful. In particular the statistics are worth mentioning as they quite intuitively play into the time trade-off.

**e)** For the aerosol retrieval on page 11, line 13-14 "The average sing scattering albedo ω and asymmetry parameter g are not subject to retrieval and are constant in all layers". What values are used?

**f)** The necessary inputs for VLIDORT are normalized as the authors state e.g. page 9 line 4-5, but this should be made clear more consistently. For instance the listed elements 4-6 on page 9 lines 9-11 should be "normalized rate of change ….". Similarly on page 12, line 10 "… what needs to be done is to calculate the normalized derivatives …" as this is what is presented in Eq. 11,12

**3)** Regarding Section 5 and error analysis:

**a)** In the description of the averaging kernels and degrees of freedom it should be noted that both are relative to the a priori information. This is especially important for the aerosol retrieval which uses Tikhonov regularization which yields an unbiased estimator contingent upon the a priori. E.g. on page 14 line 21 language similar to "DOF, the number of pieces of information independent of the a priori in the profile retrieval, …" should be used.

**b)** Section 5.3.1 is difficult to parse, particularly the first sentence: "The error originating from the cross-section is estimated by assuming that the column amount regarding to the used cross-section has a uncertainty of 3% (Wang et al., 2017)". I assume Eq. 25 has an error and should have 3% or 3.0% rather than 0.3%, otherwise I am misunderstanding. A clearer distinction in the language regarding errors in the measurements (y) as opposed to in the column or partial columns (x).

**c)** The error budget is composed in a number of different ways with some common terminology describing similar errors in the aerosol and NO$_2$ retrievals. This is relatively clear and transparent in Table 1, but can be difficult to follow in the text. For instance the measurement of error in NO$_2$ is 2.4% first quoted on page 16 line 6. Later on page 17 line 27 "measurement of noise" of 2.2% is quoted, this latter number is measurement noise in O$_4$ propagated to the NO$_2$ retrieval, a different quantity, nonetheless it can seem inconsistent. Earlier and more frequent reference to Table 1 would be useful I offer a key example:

The language at the end of Section 5.3.2 should be revised, it is difficult to understand precisely. Starting at page 17 line 27: "The propagation of the smoothing (4.6%) and measurement noise (2.2%) errors of the O4 retrieval into the NO2-retrieval results in a 5.1% error in the NO2 VCD" this appears to refer to Table 1 line 9 and is reasonably clear perhaps end the sentence here. Continuing, "while if no $O_4$-retrieval is performed successfully the error would be in our example 9.8%", here as I understand it line 7 of Table 1 is now substituted without reference to other errors, this should be stated explicitly. Finally, "In case we would include the algorithm error (7.8%) introduced by Wang et al. (2017) the error when a $O_4$-retrieval is performed successfully would be 9.4%." This is reasonably clear but there appears to be a discrepancy with line 10 of Table 1.

In Fig. 2 the $NO_2$ dSCD errors are shown, is the variability largely a reflection of the relative magnitude of the underlying dSCDs? Are the proportional errors reasonably constant areound the 2.4% value quoted in Table 1, or do they vary with viewing angle also?

**4)** Regarding the Results and Conclusions

**a)** For the limited degrees of the aerosol retrieval, the authors state (page 21 lines 2-3) that "Currently, the integration times in the spectra from which the $O_4$ dSCDs are calculated, are not long enough to ensure an $O_4$ dSCD error resulting in DOF larger than 1 for the aerosol retrieval." However, based on the error budget presented in Table 1, the measurement noise in $O_4$ is the smallest component. Should increased integration times be expected to yield significant improvement? In the next sentence: "Since we use a Tihkonov regularization for aerosol retrieval, this means that we can basically retrieve the total aerosol extinction." Based on Fig. 4(b) the retrieved DOF is approximately a total column below ~5.5km, very likely similar to AOD under most circumstances, but not necessarily the same.

**b)** Regarding the comparison with in situ $NO_2$ measurements, the authors highlight the impact of clouds on the comparison in Figs. 8 and 9 and examine the diurnal and seasonal components of the comparison in Figs. 11 and 12 respectively. Figure 10 to some degree combines all these aspects in the context of case studies. I wonder whether it is possible to build on this further. For instance, the slope of a MAX-DOAS – in situ comparison can be to some degree inferred from the information presented in Figs. 11 and 12, are there sufficient statistics to present Pearson's R on these graphs also? If so it might bring greater precision to some of the discussion. Similarly, the results in Figure 8 should have some diurnal and seasonal variation which would help point to the representativeness of the effects highlighted in the Fig. 10 case studies and accompanying discussion.

The caption to Fig. 8 says the slopes where forced to zero, while in Fig. 9 the fits have non-zero intercepts. Why the inconsistency? Does this have any significant impact?

At present it is difficult to make much of the point cloud in Fig. 9, are the correlations reasonably linear across the space? Binning data and presenting statistics might provide better insight than the present graph.

**c)** The authors conclude that the MAX-DOAS "systematically underestimates the ground level concentrations…", however, this is relative to a single in situ sensor and could in part reflect systemic persistent horizontal inhomogeneity. Such effects have been observed before e.g. (Dunlea et al., 2007; Oetjen et al., 2013; Ortega et al., 2015; Shaiganfar et al., 2011). Particularly at UNAM and in Mexico City, (Rivera et al., 2013) highlights that the MAX-DOAS at UNAM is likely to sample across a significant

horizontal gradient. This is relevant to the later discussion of future plans to compare with more sites and with satellites, especially as the Acatlán and Vallejo sites should have overlap in their sampling (Arellano et al., 2016).

Minor comments:

Page 1, line 14: "… the total error is considerably large …" large relative to what? The errors do not seem atypical, further they are quantified immediately thereafter.

Page 1, line 19-20: "it is indispensable to have the proper tools to measure them not only at ground level but also throughout the boundary layer." Consider including a citation to support that contention that boundary layer measurements are indispensable as this is quite a strong statement.

Page 2, line 5: insert "been" to have: "applications of this technique have been demonstrated to"

Page 2, line 11: change "in" to "on" for "restrictions on the usage"

Page 2, lines 12-14: this is a long sentence, consider breaking. Also consider changing "and" to "which" at the end otherwise to clarify relation of clauses, i.e. "… (MCMA) which has been …"

Page 5 line 4: "and average" here should be "an average"

Page 5 line 5: Multiple errors, homogeneity vs inhomogeneity, something can be true or untrue, consider rephrasing e.g. "… since the assumption of horizontal homogeneity likely holds less well." or " … since this likely deviates further from the assumption of horizontal homogeneity."

Page 6 line 13: eliminate double negative, perhaps replace "… is not too non-linear so …" with "… is sufficiently linear such …"

Page 7 line 6: should "unlinear" here be "non-linear"?

Page 17 line 17: "equals" should be "equal"

Page 8 line 7: References to manuscripts in preparation do not appear in the reference section. Here there is a reference to a manuscript by Wang et al. on IO whereas previously on page 6 line 10 there is a reference to a manuscript by Wang et al. on HONO. Are these two difference manuscripts or is one instance a typo?

Page 9 line 4: Here "Jacobians" is capitalized whereas it was not previously, check consistency.

Page 9 line 13: "enclosed" here should replace "inclosed" which is no longer standard.

Page 10 line 15: As described above, the temperature dependence of the cross-section is not presently implemented, as such it should likely be eliminated from Eq. 4.

Page 12 lines 14-16: There are a number of formatting errors in equations specifically, Eq. 2, 16, 22, 25. Here three equations appear but only two are numbered, specifically the normalized derivative of $\omega$ is not assigned an equation number. In Eq. 12 unlike the previous equations only the simplified expression is given, not an intermediate step in the derivation.

Page 13 lines 4,6: The logarithm in Eq. 16 is base e, since ln(x) specifically appears in the text below, these should probably match to avoid the potentially for an apparent difference.

Page 13 line 21: "produce" here should be "produces"

Page 14 line 9: "constraint" here should be "constrained"

Page 14 line 12: Should "not symmetrical" here be "asymmetric"?

Page 18 line 7: The word "most rigorous" is probably not the best choice. Depending on what the authors wish to communicate, most imposing, or least supported might be alternatives.

Page 18 line 9: eliminate the before VLIDORT

Page 19 line 7: eliminate "relatively" it is not needed.

Page 20 line 3: Here "*in situ*" appears as two word in italics which I believe is the Copernicus standard for such phrases derived from Latin; "*a priori*" should I believe appear the same way.

Page 21 line 12: eliminate "in" to get "about half of", it is not necessary

Page 23 lines 16-17: consider rephrasing sentence for clarity, perhaps "When all the coincident data is considered, regardless of if the retrieval had data available from the AERONET instrument on that day or not, the R and slope values are 0.62 and 0.39, respectively."

Page 24 line 16: "relatively" is not needed; change "despite that there are more" to "despite there being more"

Page 26 line 8: Based on Fig. 8 and the prior text, the MAX-DOAS results are on average 0.4 (or 40%) of the ground level in situ measurement. The underestimate then is the difference, namely 0.6 or (60%) is this not the case?

Page 27 line 1: I don't think the 's is needed after $NO_2$

Appendix A equations: Some of the numbers in these equations given with decimal precision are numeric factors and I don't think require the decimal precision. Some instances are the leading 1's in A8, A9, and A11, and I think all whole numbers in A16 and A20.

References: There are some formatting oddities in the references. Many but not all papers appear with both a DOI code and also a url which in many instances are redundant. The Bates citation includes a citation statistic.

References:

Arellano, J., Krüger, A., Rivera, C., Stremme, W., Friedrich, M. M., Bezanilla, A., & Grutter, M. (2016). The MAX-DOAS network in Mexico City to measure atmospheric pollutants. https://doi.org/10.20937/ATM.2016.29.02.05

Bogumil, K., Orphal, J., Homann, T., Voigt, S., Spietz, P., Fleischmann, O. ., et al. (2003). Measurements of molecular absorption spectra with the SCIAMACHY pre-flight model: instrument characterization and reference data for atmospheric remote-sensing in the 230–2380 nm region. *Journal of Photochemistry and Photobiology A: Chemistry*, *157*(2–3), 167–184. https://doi.org/10.1016/S1010-6030(03)00062-5

Burrows, J. P., Richter, A., Dehn, A., Deters, B., Himmelmann, S., Voigt, S., & Orphal, J. (1999). Atmospheric remote-sensing reference data from GOME-2. temperature-dependent absorption

cross sections of O3 in the 231–794 nm range. *Journal of Quantitative Spectroscopy and Radiative Transfer*, *61*(4), 509–517. https://doi.org/10.1016/S0022-4073(98)00037-5

Damadeo, R. P., Zawodny, J. M., Thomason, L. W., & Iyer, N. (2013). Atmospheric Measurement Techniques SAGE version 7.0 algorithm: application to SAGE II. *Atmos. Meas. Tech*, *6*, 3539–3561. https://doi.org/10.5194/amt-6-3539-2013

Dunlea, E. J., Herndon, S. C., Nelson, D. D., Volkamer, R. M., San Martini, F., Sheehy, P. M., et al. (2007). Evaluation of nitrogen dioxide chemiluminescence monitors in a polluted urban environment. *Atmospheric Chemistry and Physics*, *7*(10), 2691–2704. https://doi.org/10.5194/acp-7-2691-2007

Hermans, C., Vandaele, A. C., Carleer, M., Fally, S., Colin, R., Jenouvrier, A., et al. (1999). Absorption cross-sections of atmospheric constituents: NO2, O2, and H2O. *Environmental Science and Pollution Research*, *6*(3), 151–158. https://doi.org/10.1007/BF02987620

Oetjen, H., Baidar, S., Krotkov, N. A., Lamsal, L. N., Lechner, M., & Volkamer, R. (2013). Airborne MAX-DOAS measurements over California: Testing the NASA OMI tropospheric $NO_2$ product. *Journal of Geophysical Research: Atmospheres*, *118*(13), 7400–7413. https://doi.org/10.1002/jgrd.50550

Ortega, I., Koenig, T., Sinreich, R., Thomson, D., & Volkamer, R. (2015). The CU 2-D-MAX-DOAS instrument – Part 1: Retrieval of 3-D distributions of NO<sub>2</sub> and azimuth-dependent OVOC ratios. *Atmospheric Measurement Techniques*, *8*(6), 2371–2395. https://doi.org/10.5194/amt-8-2371-2015

Peters, E., Pinardi, G., Seyler, A., Richter, A., Wittrock, F., Bösch, T., et al. (2017). Investigating differences in DOAS retrieval codes using MAD-CAT campaign data. *Atmos. Meas. Tech*, *10*, 955–978. https://doi.org/10.5194/amt-10-955-2017

Rivera, C., Stremme, W., & Grutter, M. (2013). Nitrogen dioxide DOAS measurements from ground and space: comparison of zenith scattered sunlight ground-based measurements and OMI data in Central Mexico. *Atmósfera*, *26*(3), 401–414. https://doi.org/10.1016/S0187-6236(13)71085-3

Serdyuchenko, A., Gorshelev, V., Weber, M., Chehade, W., & Burrows, J. P. (2014). High spectral resolution ozone absorption cross-sections – Part 2: Temperature dependence. *Atmos. Meas. Tech*, *7*, 625–636. https://doi.org/10.5194/amt-7-625-2014

Shaiganfar, R., Beirle, S., Sharma, M., Chauhan, A., Singh, R. P., & Wagner, T. (2011). Estimation of NO x emissions from Delhi using Car MAX-DOAS observations and comparison with OMI satellite data. *Atmos. Chem. Phys*, *11*, 10871–10887. https://doi.org/10.5194/acp-11-10871-2011

Thalman, R., & Volkamer, R. (2013). Temperature dependent absorption cross-sections of $O_2$-$O_2$ collision pairs between 340 and 630 nm and at atmospherically relevant pressure. *Physical Chemistry Chemical Physics*, *15*(37), 15371–81. https://doi.org/10.1039/c3cp50968k

---

## Referee Comment (RC2) · Anonymous Referee #1 · 16 Jan 2019

Review of "NO2 vertical profiles and column densities from MAX-DOAS measurements in Mexico City" by Friedrich et al.

This manuscript discusses a newly developed profile retrieval code - the Mexican Max-doas Fit (MMF). Note: The first author's initials match the acronym, nicely done! The retrieval code consists of 2 parts, 1) an aerosol retrieval and 2) a trace gas retrieval using the previously retrieved aerosol profiles. This code is then used on 19 months of MAX-DOAS data measured at a location in Mexico City and the results are discussed. A comprehensive error analysis (which is great to see!) is also included in the manuscript.

It certainly is interesting to look at the complete 19 months NO2 data set (e.g. see the discussed averages of the diurnal variation) but my guess is the more interesting

studies (specially from an environmental view point) can be done by looking at individual days and using the right ancillary data to understand the NO2 variability and what causes the observed peaks.

Overall, the manuscript is well structured and the figures and table are clear and straight forward to understand. However, in some places (e.g. in Section 5, Error analysis) the text can be somewhat difficult to follow, and the manuscript could gain from having another go at streamlining the text a bit more and simplifying the structure of some of the more complicated sentences.

Specific comments:

Page 2, line 16: Replace 'giving' with 'with'

Page 2, line 17: Section 2 should be Sect. 2 for consistency, check whole manuscript

Page 2, line 19: '(constituting the forward model)' - What exactly does this mean?

Page 2, line 30: UNAM – can you please spell this out once

Page 3, Figure 1, caption. Nice overview figure. For completeness, can you please also include a brief description of the yellow and red box in the caption.

Page 3, line 3: Replace 'large' with 'long'

Page 4, line 8: Typo: 'receiving'

Page 5, line 4: Typo: 'an average'

Page 5, lines 16-25: Why was O4 not retrieved using the same wavelength interval as NO2? The much older O4 XS from Hermans et al. 1999 was used for the O4 retrieval, why not Thalman and Volkamer (2013)?

Thalman, R. and R. Volkamer, Temperature dependent absorption cross-sections of O2-O2 collision pairs between 340 and 630 nm and at atmospherically relevant pressure., Phys. Chem. Chem. Phys., 15(37), 15371–81, doi:10.1039/c3cp50968k, 2013.

Page 5, line 26: Would it be possible to say something briefly here about how the errors were determined?

Page 7: The authors explain that the retrieval code was recently updated from using the Gauss-Newton scheme to the more stable Levenberg Marquardt iteration scheme. However, this is not really relevant for the work presented here and seems to unnecessarily complicate the discussion. Unless there is a compelling reason to keep this information, I suggest to drop the relevant equations and just briefly mention in a couple of sentences (or one paragraph) that the retrieval code has been updated and how. It would also be better to have all the variables explained straight after Equation (1) and not further down the page.

Page 7, line 6: Change to 'non-linear'

Page 7, line 12: Change to 'dimension which is the number of telescope'

Page 7, line 17: Change to: 'equal to 1'

Page 7, line 23: Change to: 'for the trace gas'

Page 8, line 8: Change to: 'with the LM iteration scheme.'

Page 8, line 10: Change to: 'algorithms. For example, there are'

Page 8, line 20: Typo: 'high speed'

Page 8, line 21: 'instead of the 2x the number of layers calls'

Page 8, line 23: Jacobians always with capital J, also on Page 12 & 13

Page 8, line 23/32 and footnote: Why not refer straight to LIDORT if only that part if used anyway?

Page 8, line 32: Maybe replace with 'For each simulated atmospheric layer, '

Page 9, lines 9-14: It is not quite clear to me how the rate of change is represented in Figure 1, can you please explain . . . or I might have misunderstood?

Page 9, line 13: Should be either 'enclosed' or 'included' ?

Page 9, line 21: Replace 'is' with 'are'

Page 11, line 2: Comma needed after QRay

Page 11, line 14: 'are assumed to be constant in all layers.'

Page 11, line 15: Replace 'are' with 'is'

Page 11, line 16: Change to 'density profiles in arbitrary units from . . .'

Page 11, line 17: Change to 'heights h to provide

Page 11, line 19: I am not sure if all readers will know what is meant with an 'intensive quantity', maybe explain briefly in a footnote?

Page 13, Equation 16: Rogue bracket or is something missing?

Page 13, line 18: Gain is written in a strange font, on purpose (why?)? If not, please fix.

Page 13, line 21: 'produces'

Page 14, line 9: Add comma after fitting

Page 14, line 23: Change to 'AK matrices from the other errors.'

Page 15, line 1: 'the VMR(VMR)' – is that correct?

Page 15, lines 16/17: Why would the vertical aerosol axtinction profile not be available?

Page 16, Figure 4 caption, last sentence: 'an ideal'

Page 16, line 6: Add comma after 'operator' – makes this sentence a bit easier to read.

Page 17, Equation 25: Should that be 3% instead of 0.3%?

Page 17, line 8: Should either be 'error . . . is' or 'errors . . . are'

Page 17, line 14: Comma after retrieval

Page 17, line 15: 'contributions: a) smoothing error and b) .. error.

Page 18, line 1: Comma after (2017)

Page 18, line 3: Could use 'dependent' instead of 'not independent'.

Page 18, line 5: Delete 'it'. Comma after 'However'

Page 18, line 9: Delete 'the' before 'VLIDORT'.

Page 18, lines 10/11: Add commas after '(2017)' and 'the residual'

Page 19, Figure 6: The two solid orange lines are hard to distinguish, could use dash or dash/dot for one of them.

Page 19, Figure 6 caption: Change to 'a) The square . . .' and delete full stop after 'total'

Page 19, line 3: Better: 'errors for No2 and O4 calculated '

Page 19, line 5: Change to 'errors' and delete 'fairly'

Page 19, line 7: Delete 'relatively'

Page 19, line 8: Something is not right with this sentence & it doesn't make sense as it is written. Maybe delete 'to' or rephrase altogether.

Page 20, Table 1, caption: The last sentence is a bit hard to read; would help to add a comma after included and it needs a 'with' after better.

Page 20, line 2: Add comma after 'In this section'

Page 20, line 4: Typo: 'approx.'

Page 21, line 10: Typo: 'Currently'

Page 23, line 1: I would rather say: 'Generally, a better . . .'

[Figure]

Page 24, lines 8-11 and Figure 10: Would be really interesting to get higher resolved surface measurements as well, otherwise a small increase might be hidden in the surface data set as well. The peak only shows up clearly in the individual measurements with sufficiently high temporal resolution. Similar peak also shows up on Aug 15 and one could argue to some degree even on 9 Sep and 22 Dec with a bit of a time shift. Any idea what causes it?

Page 24, line 28: Change to: 'This might have to do with the fact that during . . .' otherwise something seems to be missing from this sentence.

Page 24, lines 20-23 & Page 26, Figure 11: Could you add a brief discussion here on the nicely (amazingly?) constant offset between surface and MAX-DOAS data, also including the uncertainties of both data sets in that discussion. Would you say that this is predominantly caused by NO2 having strong emissions on the surface which are then just diluted over the vertical range which the MAX-DOAS measurements are covering?

Page 26, line 1: Change to: 'and certain trace gases. We . . . NO2 at one . . .'

Page 27, line 1: delete ' 's'

Page 27, line 4: Add something like 'Sincs this study, it has been . . ..'

Page 27, line 10: Add 'the' before 'NO2'.

Page 31-34, References: There seems to be some doubling up of information, please check through all the references for correct formatting.

---

## Referee Comment (RC3) · Anonymous Referee #3 · 24 Jan 2019

This paper presents MMF, a new algorithm developed to retrieve vertical profiles of trace gases from MAX-DOAS measurements. The MMF code and the inversion methods implemented in it are first thoroughly described. Then MMF is applied to MAX-DOAS measurements of O4 and NO2 in Mexico City and NO2 retrieval results are compared to co-located in-situ observations. A detailed error analysis of both NO2 profile and vertical column density is also presented.

The manuscript is well written and clearly structured and the presented data set is scientifically important, especially for the validation of NO2 satellite observations in highly polluted area like Mexico City. I recommend the paper for publication in AMT after addressing the following comments and technical corrections:

General comments:

[Figure]

1/I think it would help the reader to include a map (that could be Figure no 1 of the manuscript) of the Mexico City area showing the locations of the different instruments (MAX-DOAS, AERONET, in-situ). Moreover, indicating the pointing directions of the MAX-DOAS instrument involved in this study could maybe also give some insights on the interpretation of the discrepancy between MAX-DOAS and in-situ NO2 surface concentration values. For instance, part of the underestimation of the in-situ values by the MAX-DOAS could be related to the fact that the MAX-DOAS instrument points towards a part of Mexico City which is less polluted than the location of the in-situ instrument.

In addition to the map, the location (latitude, longitude, altitude) of all the instruments should appear in the text.

2/AERONET data are used as input in the retrieval but also as ancillary data for the sky conditions screening. Was there any attempt to compare the retrieved AODs with those from AERONET ? It can be a good check for the aerosol retrieval part of the profiling. Also related: it seems that the availability of AERONET observations has been used as a quality control (QC) flagging for the MAXDOAS retrievals. Was there any attempt to apply a QC flagging which is more specific to the MAX-DOAS retrievals, e.g. using parameters like DOF and the RMS of the differences between measured and calculated dSCDs ?

Specific comments:

1/Page 4, beginning of Section 2.3: It would be good to list the exact elevation angle values of a typical scan.

2/According to Section 3, it seems that aerosol profile retrieval is done in the UV range and then retrieved profiles are used as input for the NO2 profile retrieval in the visible range. Has a correction been applied to the retrieved extinction/AOD for taking into account for the wavelength dependence of the AOD/extinction ? If not, then this approximation should be included as an additional error source in Table 1.

3/Page 6, lines 8-11: Maybe you could add a couple of sentences about the performance of MMF in these profile comparison exercises. Please note that in the meantime, Friess et al. is now published in AMTD.

4/Page 8, line 7: Is it IO or HONO (cf page 6, end of Sect. 4) ?

5/Page 8, Sect. 4.2: You should add a paragraph on the SCIATRAN RTM, which has been also used in past MAX-DOAS profiling studies (see e.g. Friess et al., AMTD, 2018).

6/Page 10, line 1: what type of interpolation is done for the pressure, temperature profiles ?

7/Page 10, line 14: a correlation length of 500m is used. Did you perform sensitivity tests on this parameter in order to estimate its impact on the retrieved profiles and on the level of agreement with in-situ measurements ?

8/Page 19, Figure 6b: A priori profile should be also included in this Figure in order to see how far the retrieved profile differs from the a priori one.

9/Page 26, lines 5-10: According to the authors, a possible reason for the underestimation of the in-situ surface concentration by the MAX-DOAS is the fact that the MAX-DOAS instrument has a maximum sensitivity around 1km and less sensitivity close to the ground. This feature is quite unusual since normally lowest elevation angles have a higher weight in the retrieval due to higher AMFs, and therefore MAX-DOAS measurements close to the ground. Could the authors elaborate on that ? Another possible explanation for the discrepancy is that, related to the horizontal extent and the pointing direction of the MAX-DOAS measurements, both instruments probe different air masses. I think this point should be also added in the discussion.

10/Acknowledgements: Depending on the conditions of use, the sources of ancillary data included in your study should be acknowledged here.

Technical corrections:

'->' denotes 'should be replaced by'

1/Page 1, line 1: '. . .to retrieve profiles. . .' -> '. . .to retrieve vertical profiles. . .'

2/Page 1, line 10: '. . .at the Universidad Nacional Autónoma de México (UNAM) campus.'

3/Page 1, line 20: '. . .The Multi-AXis Differential Optical Absorption Spectroscopy (MAX-DOAS) technique. . ..'

4/Page 2, line 30: '. . .at the Universidad Nacional Autónoma de México (UNAM).'

5/Page 6, line 5: 'Sect.4.1' -> 'Sect. 4.1'. This typo should be corrected throughout the manuscript; similar corrections also needed for 'Fig.' and 'Eq.'.

6/Page 6, line 9: 'CINDI2' -> 'CINDI-2'

7/Page 10, line 14: '. . .Eq. 4:'; same on Page 11, line 4

8/Page 11, line 20: 'extincion' -> 'extinction'

9/Page 16, legend of Fig. 4: (b,right)' -> '(b, right)'; should be also corrected for (a, left).

10/Page 19, legend of Fig. 6: 'total error'

11/Page 19, line 3: 'algoritm' -> 'algorithm'

12/Page 20, line 4: 'aprox.' -> 'approx.'

13/Page 21, line 11: 'Curretly' -> 'Currently'

14/Page 24, end of line 23: A reference could be added here.
* * *

---

## Author Comment (AC1) · 4 Feb 2019

We thank the referee for his very careful review, and his constructive suggestions. In the following, we answer his specific questions. In oder to facilitate the reference to the questions and proposed changes, we use the following color coding:

**Color coding:**
 Reviewer comment
 Our comment
 Suggested changes in the manuscript
* * *
**1)** Regarding the dSCD retrieval and Section 3:
**a)** The authors use a zenith measurement prior to the scan to analyze the scan. If the upper atmospheric contribution to the dSCDs changes during the scan this can lead to signal in the measurements which is from the upper atmosphere being falsely attributed to lower altitudes especially in low eastward elevation angles at the end of the scan. The effect would be expected to lead to lower VCDs in the morning and higher VCDs in the evening, especially in winter. Because of the short 7 minute scan time this effect would likely only be significant at twilight. Does the instrument acquire data at twilight which are included in the analysis? Can the authors bound the impact of such an effect and compare it to the magnitude of their error budget?

Currently we are not considering any data measured at twilight in the analysis and hence the effect of geometry should be small within a 7 min time window. The difference for the zenith measurements is likely to reflect the difference in atmospheric conditions over 7 minutes. If these differences are huge, one of the assumptions made in the concept of MAXDOAS, same conditions for measurements in all off-axis directions, is violated and the scan in question most likely not suitable for profile retrieval.
A way of estimating the error from using the same reference would be to look at the difference between two zenith dscds of two consecutive scans using the noon zenith sky as reference. We made such a test and found that typically, these differences are of the order of several $10^{41.}$ molec$^2$/cm5. This is smaller/ of the order of the typical fitting errors for O4 of typically a couple 10^42.

b) For the fitting setting in the retrieval the authors use older cross-sections where newer cross-sections for the same gases are increasingly standard in the community e.g. (Damadeo et al., 2013; Peters et al., 2017). For O3 they use (Burrows et al., 1999) rather than (Bogumil et al., 2003) or (Serdyuchenko et al., 2014) and for O4 they use (Hermans et al., 1999) rather than (Thalman & Volkamer, 2013). Was there any particular reason for these choices? Is it based on Orphal, 2002 cited later?

There was no particular reason for using the chosen cross-sections, they were chosen because they are quite standardly used for retrievals. At the moment, we are re-running all our dataset using fitting settings as in Peters et al. (2017). A comparison among our chosen settings and the Peters et al., (2017) settings is shown in Fig.1 (data for one scan sequence).
a) Our retrieval settings.
b) Peters et al., 2017 retrieval settings.
c) a and b plotted together

d) Difference (a-b) with the error bars: $\sqrt{\varepsilon_a^2 + \varepsilon_b^2}$

From this graphs it is evident that the difference arising from the use of different cross-sections is small compared to the typical fitting error in the DOAS retrieval.

[Figure]

Fig1: Comparison of results for one scan  using retrieval settings as used in the manuscript with results using retrieval settings as in Peters et al. (2017)

**2)** Regarding the profile retrieval method description and Section 4

**a)** At present the aerosol and trace gas inversions in Fig. 1 are presented as the same, whereas the former uses Tikhonov regularization and the latter optimal estimation. This should be reflected in the figure as it is in the text.

We agree with the reviewer and would like to make the following change in Figure 1. We include a suggested Figure to replace Figure 1 in the manuscript here as Fig. 2.

\* Replace the first red box by: "Inversion using Thikonov Regularization"

\* Add on the right hand side of that box a box with "L1 Scaling fator" and connect this box with a left-ward pointing arrow to the red box

\* Replace the second red box by "Inversion using OE"

\* Add on the right hand side of this box a box with "Sa  from  simulations"

Also, reviewer 1 has pointed out a missing "rates of change" as input quantities, we also add this to the orange and green box that leads to the yellow VLIDORT boxes.

[Figure]

Fig2 : replaces Figure 1 in manuscript

**b)  T**he code used to analyze the 18 month data set presented in the work utilized a Gauss-Newton (GN) iteration scheme for inversion, however, MMF has since been updated to utilize a Levenberg Marquardt (LM) iteration scheme, as well as other more minor updates. At present both schemes are described somewhat in parallel, and the authors are diligent in describing which scheme they are discussing. Nonetheless, equations for the GN scheme, which was used, are sometimes left out in favor of the more current LM scheme equivalents, leaving the methods applied not fully transparent to the reader. I would

recommend describing the GN scheme as default as it is most relevant to the titular topic of the work and collecting and describing the changes for the LM scheme either all together in a dedicated section or within the relevant subsections.

We agree with the reviewer that it is not very clear and we intent to improve this by moving the changes to an appendix and concentrate on the description of the actual used code in the main text. Specifically, this means:

* moving the mentioning of the change to LM to a footnote, which then refers to a new appendix A

* Creating an Appendix A (Current appendix A will then change to appendix B). The current footnote 1 will then be part of that appendix A. Eq. 2 and the cost-function (Eq. 3) Is then moved to that appendix.

* The 3rd paragraph of Sect. 4 lines 8--10 ["MMF has been participating...] is moved to the Appendix as well.

* Last paragraph of Sect. 4.1 (page 8, lines 4–8) is moved (and slightly reformulated) to the new Appendix A

However, we are not sure which equations the author sees missing.

Page 6, line 8 –11 replaced by:

"The retrieval time per aerosol and trace gas retrieval with the Mexico City set-up is roughly half a minute for each scan, but highly dependent on the conditions."

Sect. 4.1 Inversion theory  replaced by (This also includes a change addressing point c):

"The inversion strategy relies on the fact that the problem is sufficiently linear so that in the iteration procedure, the new value for the quantity vector in question x (either the aerosol total extinction per layer or the trace gas optical depth per layer) can be calculated using a Gauss-Newton (GN) scheme[1] according to Eq.1 (Rodgers, 2000). This step corresponds to the red box and arrows in Fig. 1

[Eq. 1]

Here,  superscript T denotes transposed, superscript −1 denotes the inverse. The index i is the iteration index, the subscript a indicates a-priori values. $S_m$ is the measurement error covariance matrix, y denotes the vector of measured differential slant column densities. $F(x_i)$ are the simulated differential slant column densities, calculated using the forward model with input profile $x_i$ . Both y and $F(x_i)$ are vectors of dimension (# telescope viewing angles). $K_i = \partial F(x)_l /\partial x_n$ is the jacobian matrix at the i-th iteration describing the change of simulated dSCD for viewing angle l when the profile x in layeris varied.

In the case of optimal estimation (OE), the regularization matrix R is equal to the inverse of the a-priori covariance matrix, $R = S_a^{-1}$ . OE regularization is used for trace gas retrieval.  Other regularization matrices are possible, see e.g. Steck (2002).

For the aerosol retrieval used in this study, we use the L1 operator ($R = L1^T \alpha L1$) where the scaling parameter α is set to a constant value of 20 and is supplied via an input script to limit the degrees of

freedom (DOF) to just slightly above 1. Different scalings for the upper layers and lower layers could be supplied, as well as a complete regularization matrix R.

New footnote (1):

In a recent update, the GN scheme was replaced by the more stable Levenberg Marquardt (LM) iteration scheme, more details on recent changes can be found in Appendix A.

New appendix A: Recent updates of the code

In a recent update of the code, implemented after the analysis presented here (i.e. not used for obtaining the results here) the retrieval space was changed from linear space to logarithmic retrieval space. This means that the retrieval works in a linear (dscd measurement)-logarithmic(profile retrieval) space now. This enhances the nonlinearity of the problem and required a change of iteration scheme.

The GN iteration scheme (Eq. 1), was replaced by a slightly slower but more stable Levenberg Marquardt (LM) iteration scheme (Rodgers, 2000):

[Equation A1, Former Eq. 2]

In order to counteract the slowdown of the retrieval, more restrictions were placed on the observation geometry for a single scan: a single relative azimuth angle and a single solar zenith angle per scan. This means in particular that two different viewing directions cannot be treated as a single scan any longer. Although this means a significant cut in flexibility, it results in a retrieval time speed up of a factor of 4 and a more typical retrieval time per scan (for each component) is around 5 seconds.

Tests using the logarithm of the partial layer vertical column density (for NO2 retrieval) or layer extinction profile (for aerosol retrieval) motivated the change to the LM iteration scheme due to the increased non-linearity when working in a semi-log space as state-measurement space.

With this new configuration, MMF has been participating in the Round-Robin comparison of different retrieval codes for the FRM4DOAS project (Frieß et al., in preparation). It has also participated in the profile retrieval from dSCD from the CINDI2 campaign, both for NO2 and HCHO (Tirpitz et al., in preparation) as well as for HONO (Wang et al., in preparation).

The LM scheme of Eq. 2 has currently only been tested with OE and not with Thikonov regularization, i.e. the aerosol retrieval was also performed using OE.

**c)** At the top of page 8 is the following paragraph: "For the aerosol retrieval used in this study, we use the L1 operator (R = L1 T α L1) where the scaling parameter α is supplied via an input script to limit the degrees of freedom (DOF) to just slightly above 1. Different scalings for the upper layers and lower layers can be supplied, as well as a complete regularization matrix R." I understand the latter sentence to describe a capability of MMF, but how was the regularization conducted for the analysis presented later? Was a constant α determined such that the DOF was just over 1 or was something else done?

We used a  constant scaling of 20 for all layers. This ensured an average dof slightly larger than 1. We would

like to add this in the text, see change suggestion to (b), or copied here below:

"For the aerosol retrieval used in this study, we use the L1 operator (R = L1T αL1) where the scaling parameter α is set to a constant value of 20 and is supplied via an input script to limit the degrees of freedom (DOF) to just slightly above 1. Different scalings for the upper layers and lower layers could be supplied, as well as a complete regularization matrix R."

**d)** Discussing the advantages of MC RTM codes, the ability to model statistically rare photons and output information of the distribution of photons is also useful. In particular the statistics are worth mentioning as they quite intuitively play into the time trade-off.

We agree with the reviewer, that the discussion of MC is very interesting, however we prefer not to include that discussion here. It is not really the topic of this paper to give an extended review on RT codes not used in the retrieval technique presented here. We do mention the greater accuracy of MC codes though. We are no experts in MC codes and hence would also not feel very comfortable to discuss them in great detail.

**e)** For the aerosol retrieval on page 11, line 13-14 "The average sing scattering albedo ω and asymmetryparameter g are not subject to retrieval and are constant in all layers". What values are used?

We use values from Aeronet as is mentioned on the same page in lines 9–11. This was not clear, we will move the definition of g and ω to that paragraph and mention more explicitly that we used extra-/interpolations:

Replace lines 9 – 12 on page 11 (first paragraph of Sect. 4.2.2) by:

"The (a-priori) aerosol data for total optical depth, average single scattering albedo ω and asymmetry parameter g (used to calculate the phase function moments) are time interpolated values from the co-located AERONET (Aerosol Robotic Network) station in Mexico City (V2, level 1.5 at http://aeronet.gsfc.nasa.gov). They are also extra-/interpolated at the retrieval wavelength. The a-priori shape of the profile is taken from hourly averaged ceilometer data (García-Franco et al., 2018), interpolated at the middle layer height h of each layer."

**f)** The necessary inputs for VLIDORT are normalized as the authors state e.g. page 9 line 4-5, but this should be made clear more consistently. For instance the listed elements 4-6 on page 9 lines 9-11 should be "normalized rate of change ....". Similarly on page 12, line 10 "... what needs to be done is to calculate the normalized derivatives ..." as this is what is presented in Eq. 11,12

We agree with the reviewer that this is not consistently written. We will implement the changes suggested by the reviewer and also make the following change:

Page 12, line 18: "For the trace gas jacobian calculation, the corresponding normalized derivatives are:"

**3)** Regarding Section 5 and error analysis:

**a)** In the description of the averaging kernels and degrees of freedom it should be noted that both are

relative to the a priori information. This is especially important for the aerosol retrieval which uses Tikhonov regularization which yields an unbiased estimator contingent upon the a priori. E.g. on page 14 line 21 language similar to should be used.

We agree with the reviewer and thank him for his suggestion, which we will use in the revised manuscript:

"DOF, the number of pieces of information independent of the a priori in the profile retrieval, …"

**b)** Section 5.3.1 is difficult to parse, particularly the first sentence: "The error originating from the cross-section is estimated by assuming that the column amount regarding to the used cross-section has a uncertainty of 3% (Wang et al., 2017)". I assume Eq. 25 has an error and should have 3% or 3.0% rather than 0.3%, otherwise I am misunderstanding. A clearer distinction in the language regarding errors in the measurements (y) as opposed to in the column or partial columns (x).

The reviewer is correct: it should be 3.0% in equation 25.

We agree also that the first sentence is confusing and therefore we change it to : "The error originating from an uncertainty in the cross-section of 3% (Wang et al., 2017) is also around 3.0% in the vertical column" and similar in the lower profile.

The reviewer well understood, that the profile shows smaller spectroscopic errors where it is dominated by the a priori information.

**c)** The error budget is composed in a number of different ways with some common terminology describing similar errors in the aerosol and NO2 retrievals. This is relatively clear and transparent in Table 1, but can be difficult to follow in the text.

 For instance the measurement of error in NO2 is 2.4% first quoted on page 16 line 6. Later on page 17 line 27 "measurement of noise" of 2.2% is quoted, this latter number is measurement noise in O4 propagated to the NO2 retrieval, a different quantity, nonetheless it can seem inconsistent.

Earlier and more frequent reference to Table 1 would be useful I offer a key example:

The language at the end of Section 5.3.2 should be revised, it is difficult to understand precisely. Starting at page 17 line 27:

"The propagation of the smoothing (4.6%) and measurement noise (2.2%) errors of the O4 retrieval into the NO2-retrieval results in a 5.1% error in the NO2 VCD" this appears to refer to Table 1 line 9 and is reasonably clear perhaps end the sentence here. Continuing, "while if no O4- retrieval is performed successfully the error would be in our example 9.8%", here as I understand it line 7 of Table 1 is now substituted without reference to other errors, this should be stated explicitly.

Finally, "In case we would include the algorithm error (7.8%) introduced by Wang et al. (2017) the error when a O4-retrieval is performed successfully would be 9.4%." This is reasonably clear but there appears to be a discrepancy with line 10 of Table 1.

The referee is correct, there is a mistake: the error is 9.318% -> 9.3 %

The error when a O4-retrieval is performed successfully would be **9.3%**. However the algorithm-error is calculated from the resulting residual of the fit and is not independent on the other error sources as mentioned earlier.

In the revised manuscript we also will refer to table 1 as son as possible as the reviewer suggested.

In Fig. 2 the NO2 dSCD errors are shown, is the variability largely a reflection of the relative magnitude of the underlying dSCDs? Are the proportional errors reasonably constant areound the 2.4% value quoted in Table 1, or do they vary with viewing angle also?

The referee makes a very good point: The 2.4% is just the average, and the error is not constant in percentage. For high elevation angles, the dSCD can take values of 0 or even below 0. Hence, to express the error in terms of percentage for low elevation angles is a bit tricky. In order to give an idea of the dependence of the error in terms of percentage for different elevation angles (i.e. like Fig. 2 in the manuscript but in percentage instead of absolute errors), we calculate the percentage, but w.r.t the average dscd at that elevation angle instead of the (sometime negative) actual dscd:

[Figure]

Fig. 3: As Figure 2 in manuscript but in terms of percentage w.r.t average dscd.

4) Regarding the Results and Conclusions

a) For the limited degrees of the aerosol retrieval, the authors state (page 21 lines 2-3) that "Currently, the integration times in the spectra from which the O4 dSCDs are calculated, are not long enough to ensure an O4 dSCD error resulting in DOF larger than 1 for the aerosol retrieval." However, based on the error budget presented in Table 1, the measurement noise in O4 is the smallest component. Should increased integration times be expected to yield significant improvement? In the next sentence: "Since we use a Tihkonov regularization for aerosol retrieval, this means that we can basically retrieve the total aerosol extinction." Based on Fig. 4(b) the retrieved DOF is approximately a total column below ~5.5km, very likely similar to AOD under most circumstances, but not necessarily the same.

The statement "Currently, the integration times in the spectra from which the O4 dSCDs are calculated, are not long enough to ensure an O4 dSCD error resulting in DOF larger than 1 for the aerosol retrieval" refers to the limited DOF that could be achieved from a profile retrieval due to the large (typically of the order of a couple of $10^{42}$ molec$^2$/ cm$^5$) dscd error. The reviewer also cites our sentence "Since we use a Tihkonov regularization for aerosol retrieval, this means that we can basically retrieve the total aerosol extinction". This means that we retrieve the total column but likely not the correct profile if this is hugely different from the a priori profile. We would like to draw the attention of the reviewer to page 15 line 20: "The a priori information about the optical properties described by the aerosol extinction profile is designed for cloud free days and therefore the error analysis is just valid for such cloud free days". This means that the errors estimated only hold for those days were the true profile shape is close to the assumed a-priori shape. A 100% error variance was assumed in the error estimation.

b) Regarding the comparison with in situ NO2 measurements, the authors highlight the impact of clouds on the comparison in Figs. 8 and 9 and examine the diurnal and seasonal components of the comparison in Figs. 11 and 12 respectively. Figure 10 to some degree combines all these aspects in the context of case studies. I wonder whether it is possible to build on this further. For instance, the slope of a MAX-DOAS – in situ comparison can be to some degree inferred from the information presented in Figs. 11 and 12, are there sufficient statistics to present Pearson's R on these graphs also? If so it might bring greater precision to some of the discussion. Similarly, the results in Figure 8 should have some diurnal and seasonal variation which would help point to the representativeness of the effects highlighted in the Fig. 10 case studies and accompanying discussion.

We agree completely that looking at the correlation coefficients of individual days will give further insight into understanding in more depth the effects of the local dynamics and vertical and horizontal inhomogeneities. This will be interesting even more so when the surface in situ data is analyzed with a higher temporal resolution (currently we had only the hourly mean data available from the monitoring station which does limit the calculation of a reliable Pearson's coefficient). However, we believe that these comparisons succeed in the general objective sought of this study which is to show that the MAX-DOAS results for the lower layers follow reasonably well what is being measured at the surface with a more conventional methodology. We do have the intention of using our data in future investigations to study specific events and understand the individual characteristic that each instrument is capturing depending on their location within the city.

The caption to Fig. 8 says the slopes where forced to zero, while in Fig. 9 the fits have non-zero intercepts. Why the inconsistency? Does this have any significant impact? At present it is difficult to make much of the point cloud in Fig. 9, are the correlations reasonably linear across the space? Binning data and presenting statistics might provide better insight than the present graph.

The slopes in the fits for Fig. 8 were forced to zero deliberately in order to have a robust way to capture the changes in the slope as the number of layers was increased. As can be seen in the offsets reported in Fig. 9, -3.4 and 0.1 ppb, y-intercepts are small for both data sets and will produce insignificantly small changes in the slopes in case they would also be forced to zero. The purpose of this figure was to highlight how the correlation is affected in cloud (R=54) vs. clear-sky (R=74) conditions, and the red and blue solid straight lines are clearly depicting the change in both data sets. We consider replacing Fig. 9 with one where the intercept is also forced to 0 in accordance to Fig.8. This would be Fig.4 below:

[Figure]

Fig. 4: As Figure 9 in manuscript but with forced offset 0.

Regarding the binning, we don't think binning the data will help making this distinction clearer.

c) The authors conclude that the MAX-DOAS "systematically underestimates the ground level concentrations…", however, this is relative to a single in situ sensor and could in part reflect systemic persistent horizontal inhomogeneity. Such effects have been observed before e.g. (Dunlea et al., 2007; Oetjen et al., 2013; Ortega et al., 2015; Shaiganfar et al., 2011). Particularly at UNAM and in Mexico City, (Rivera et al., 2013) highlights that the MAX-DOAS at UNAM is likely to sample across a significant horizontal gradient. This is relevant to the later discussion of future plans to compare with more sites and with satellites, especially as the Acatlán and Vallejo sites should have overlap in their sampling (Arellano et al., 2016).

We fully agree with these statements, and as mentioned before, this will be investigated further in future work. For example, we are exploring the differences in VCD's obtained when MMF analyses distinctively data measured from easterly or westerly scans. The horizontal gradients are evident already from satellite data as has been accurately pointed out by the reviewer.

**M**inor comments:

1. Page 1, line 14: "... the total error is considerably large ..." large relative to what? The errors do not seem atypical, further they are quantified immediately thereafter.

Change to "The total error, depending on the exact counting is 14 –20 % and this work provides new and relevant information about NO2 in the boundary layer of Mexico City."

2. Page 1, line 19-20: "it is indispensable to have the proper tools to measure them not only at ground level but also throughout the boundary layer." Consider including a citation to support that contention that boundary layer measurements are indispensable as this is quite a strong statement.

Suggested reference is Franco-García et al., 2018.

3. Page 2, line 5: insert "been" to have: "applications of this technique have been demonstrated to"

corrected

4. Page 2, line 11: change "in" to "on" for "restrictions on the usage"

corrected

5. Page 2, lines 12-14: this is a long sentence, consider breaking. Also consider changing "and" to "which" at the end otherwise to clarify relation of clauses, i.e. "... (MCMA) which has been ..."

We will follow the suggestions

6. Page 5 line 4: "and average" here should be "an average"

corrected

7. Page 5 line 5: Multiple errors, homogeneity vs inhomogeneity, something can be true or untrue, consider rephrasing e.g. "... since the assumption of horizontal homogeneity likely holds less well." or "... since this likely deviates further from the assumption of horizontal homogeneity."

Thank you for your suggestions. We will implement the latter.

8. Page 6 line 13: eliminate double negative, perhaps replace "... is not too non-linear so ..." with "... is sufficiently linear such ..."

We will adapt the suggestion.

9. Page 7 line 6: should "unlinear" here be "non-linear"?

corrected

10. Page 7 line 17: "equals" should be "equal"

corrected

11. Page 8 line 7: References to manuscripts in preparation do not appear in the reference section. Here there is a reference to a manuscript by Wang et al. on IO whereas previously on page 6 line 10 there is a reference to a manuscript by Wang et al. on HONO. Are these two difference manuscripts or is one instance a typo?

It is a typo, it should be HONO. Thank you! We will include the manuscripts in preparation to the reference section.

12. Page 9 line 4: Here "Jacobians" is capitalized whereas it was not previously, check consistency.

Jacobian everywhere now.

13. Page 9 line 13: "enclosed" here should replace "inclosed" which is no longer standard.

corrected

14. Page 10 line 15: As described above, the temperature dependence of the cross-section is not presently implemented, as such it should likely be eliminated from Eq. 4.

That is correct, temperature dependence from Eq.4 removed

15. Page 12 lines 14-16: There are a number of formatting errors in equations specifically, Eq. 2, 16, 22, 25. Here three equations appear but only two are numbered, specifically the normalized derivative of $\omega$ is not assigned an equation number. In Eq. 12 unlike the previous equations only the simplified expression is given, not an intermediate step in the derivation.

Eq.2: removed closing square bracket after superscript -1 (This Eq. will be moved to a new Appendix A as outlined above)

Eq. 11 – 12: This indeed should be 3 numbers for three equations, this will be fixed.

Eq. 12: That is correct, we skipped it because it was not very readable and we felt that it did not add to the understanding. However, we can of course add it if the reviewer thinks that it adds to the understanding.

Eq. 16: Removed last opening round bracket and changed log to ln.

Eq.22: VMR moved to subscript

Eq.25:  Changed   $0.3^2$  -> $3.0^2$

16. Page 13 lines 4,6: The logarithm in Eq. 16 is base e, since ln(x) specifically appears in the text below, these should probably match to avoid the potentially for an apparent difference.

This is a good point, this will be changed, see comment to 15 above.

17. Page 13 line 21: "produce" here should be "produces"

corrected

18. Page 14 line 9: "constraint" here should be "constrained"

corrected

19. Page 14 line 12: Should "not symmetrical" here be "asymmetric"?

corrected

20. Page 18 line 7: The word "most rigorous" is probably not the best choice. Depending on what the authors wish to communicate, most imposing, or least supported might be alternatives.

We take "least supported" then.

21. Page 18 line 9: eliminate the before VLIDORT

corrected

22. Page 19 line 7: eliminate "relatively" it is not needed.

corrected

23. Page 20 line 3: Here "in situ" appears as two word in italics which I believe is the Copernicus standard for such phrases derived from Latin; "a priori" should I believe appear the same way.

We changed all "a-priori" to italic "a priori".

24. Page 21 line 12: eliminate "in" to get "about half of", it is not necessary

we changed this to "for about half of the coincident measurements there was…" (i.e. removed "in" and added "for" before about".

25. Page 23 lines 16-17: consider rephrasing sentence for clarity, perhaps "When all the coincident data is considered, regardless of if the retrieval had data available from the AERONET instrument on that day or not, the R and slope values are 0.62 and 0.39, respectively."

Thank you for the suggestion which we are happy to use.

26. Page 24 line 16: "relatively" is not needed; change "despite that there are more" to "despite there being more"

corrected

27. Page 26 line 8: Based on Fig. 8 and the prior text, the MAX-DOAS results are on average 0.4 (or 40%) of the ground level in situ measurement. The underestimate then is the difference, namely 0.6 or (60%) is this not the case?

Agreed, changed to:

"However, the MAX-DOAS systematically underestimates the ground level concentrations by a factor of about 0.6. "

28. Page 27 line 1: I don't think the 's is needed after NO2

corrected

29. Appendix A equations: Some of the numbers in these equations given with decimal precision are numeric factors and I don't think require the decimal precision. Some instances are the leading 1's in A8, A9, and A11, and I think all whole numbers in A16 and A20.

Will be modified.

30. References: There are some formatting oddities in the references. Many but not all papers appear with both a DOI code and also a url which in many instances are redundant. The Bates citation includes a citation statistic.

The references were checked, we will only keep the url  and modify this  in the file.

---

## Author Comment (AC2) · 4 Feb 2019

Answer to Referee 1 (2nd referee report received)

We would like to thank the reviewer for his comments. We agree with the referee that the text was a bit hard to follow in places and we hope that we could improve this by following both his and the other reviewers advices and suggestions for improvement. Since we received a comment from reviewer two before we received a comment from reviewer one, we are sometimes referring in this answer to the answer to reviewer 2.
Below, we comment on the first reviewers specific comments. For easier reference, we added a number to each comment. We use the following color coding:

Color coding:
     reviewer comment
     our answer
     proposed change in manuscript
* * *
Review of "NO2 vertical profiles and column densities from MAX-DOAS measurements in Mexico City" by Friedrich et al.
This manuscript discusses a newly developed profile retrieval code - the Mexican Max-doas Fit (MMF). Note: The first author's initials match the acronym, nicely done! The retrieval code consists of 2 parts, 1) an aerosol retrieval and 2) a trace gas retrieval using the previously retrieved aerosol profiles. This code is then used on 19 months of MAX-DOAS data measured at a location in Mexico City and the results are discussed. A comprehensive error analysis (which is great to see!) is also included in the manuscript.
It certainly is interesting to look at the complete 19 months NO2 data set (e.g. see the discussed averages of the diurnal variation) but my guess is the more interesting studies (specially from an environmental view point) can be done by looking at individual days and using the right ancillary data to understand the NO2 variability and what causes the observed peaks.
Overall, the manuscript is well structured and the figures and table are clear and straight forward to understand. However, in some places (e.g. in Section 5, Error analysis) the text can be somewhat difficult to follow, and the manuscript could gain from having another go at streamlining the text a bit more and simplifying the structure of some of the more complicated sentences.

1) Page 2, line 16: Replace 'giving' with 'with'
corrected
2) Page 2, line 17: Section 2 should be Sect. 2 for consistency, check whole manuscript
checked in whole manuscript, thank you!
3) Page 2, line 19: '(constituting the forward model)' - What exactly does this mean?
This means that the forward model in our case is a radiative transfer code and that we talk about radiative transfer codes here, because our forward model is a radiative transfer model, more specifically, VLIDORT.
4) Page 2, line 30: UNAM – can you please spell this out once
done at first appearance now, i.e. at the beginning of Sect. 2
5) Page 3, Figure 1, caption. Nice overview figure. For completeness, can you please also include a brief description of the yellow and red box in the caption.
Yes. As response to reviewer 2, we also made small adjustments to the figure, see answer to referee 2. The new caption is describing that version of the figure. We added to the caption:

"The yellow boxes represent the forward modelling steps. The red boxes are the inversion steps, using Thikonov regularization for aerosol retrieval and optical estimation (OE) for tracegas retrieval."

As a response to (23) below, we also now added "& rates of change" in the orange and green boxes before the yellow "VLIDORT box".

6) Page 3, line 3: Replace 'large' with 'long'

corrected

7) Page 4, line 8: Typo: 'receiving'

corrected

8) Page 5, line 4: Typo: 'an average'

corrected

9) Page 5, lines 16-25: Why was O4 not retrieved using the same wavelength interval as NO2? The much older O4 XS from Hermans et al. 1999 was used for the O4 retrieval, why not Thalman and Volkamer (2013)?
Thalman, R. and R. Volkamer, Temperature dependent absorption cross-sections of O2-O2 collision pairs between 340 and 630 nm and at atmospherically relevant pressure., Phys. Chem. Chem. Phys., 15(37), 15371–81, doi:10.1039/c3cp50968k, 2013.

The windows for NO2 and O4 dscd fitting are chosen to enclose pronounced absorption lines for the species in question and are widely used windows. We agree with the referee that there are O4 windows that are closer to the chosen NO2 window (405 -- 465 nm) than our choice (336 -- 390 nm) and that it had been a better choice to use one of those (e.g. 450 -- 520 nm). However, the difference in middle wavelength had only been 22 nm less (i.e. +50 instead of -72). We would also like to refer to our answer to reviewer 3 question 2 regarding correction for aod.  Regarding the choice of cross-sections, there was no specific reason for the choice of cross-section.  We would like to refer the reviewer 1 to answer 1b to the mayor comments from reviewer 2, where we also include a test for changing the retrieval settings. Our main finding is that the effect of changing the cross-sections is small.

10) Page 5, line 26: Would it be possible to say something briefly here about how the errors were determined?

The dscd error is calculated directly within qdoas. We use this error as dscd error without any modification or addition. We refer the reviewer to pages 28 -- 29 ("Errors on Slant Column Densities) of the qdoas manual (http://uv-vis.aeronomie.be/software/QDOAS/QDOAS_manual.pdf) for details on the dscd error calculation within qdoas.

11) Page 7: The authors explain that the retrieval code was recently updated from using the Gauss-Newton scheme to the more stable Levenberg Marquardt iteration scheme. However, this is not really relevant for the work presented here and seems to unnecessarily complicate the discussion. Unless there is a compelling reason to keep this information, I suggest to drop the relevant equations and just briefly mention in a couple of sentences (or one paragraph) that the retrieval code has been updated and how. It would also be better to have all the variables explained straight after Equation (1) and not further down the page.

We fully agree with the referee. Referee 2 (first report received) had a similar comment. We moved all explanations regarding changes to the code  into an appendix. This also leads to all symbols in Eq. 1 being defined right after its appearance.  For details on the changes, please see the answer 2b to the major comments section from referee report from reviewer 2.

12) Page 7, line 6: Change to 'non-linear'

corrected

13) Page 7, line 12: Change to 'dimension which is the number of telescope'

done

14) Page 7, line 17: Change to: 'equal to 1'

corrected

15) Page 7, line 23: Change to: 'for the trace gas'

included "the"
16) Page 8, line 8: Change to: 'with the LM iteration scheme.'
included "the"
17) Page 8, line 10: Change to: 'algorithms. For example, there are'
Added "For example"
18) Page 8, line 20: Typo: 'high speed'
corrected
19) Page 8, line 21: 'instead of the 2x the number of layers calls'
corrected
20) Page 8, line 23: Jacobians always with capital J, also on Page 12 & 13
now capitalized everywhere.
21) Page 8, line 23/32 and footnote: Why not refer straight to LIDORT if only that part if used anyway?
VLIDORT and LIDORT are actually different code packages with different version numbers. Since it might be that they, of course by accident, include different "features" (i.e. bugs), we think that it is more accurate to state exactly which code and which version was used.
22) Page 8, line 32: Maybe replace with 'For each simulated atmospheric layer, '
corrected
23) Page 9, lines 9-14: It is not quite clear to me how the rate of change is represented in Figure 1, can you please explain . . . or I might have misunderstood?
This was perhaps not clear. We did not refer specifically to the rate of change. We removed the sentence. However, we also realized that the rate of change as layer input was indeed missing in the diagram. We added "& rates of change" in the orange and green box before each of the yellow VLIDORT boxes.
24) Page 9, line 13: Should be either 'enclosed' or 'included' ?
enclosed
25) Page 9, line 21: Replace 'is' with 'are'
corrected
26) Page 11, line 2: Comma needed after Qray
corrected
27) Page 11, line 14: 'are assumed to be constant in all layers.'
corrected
28) Page 11, line 15: Replace 'are' with 'is'
corrected
29) Page 11, line 16: Change to 'density profiles in arbitrary units from . . .'
corrected
30) Page 11, line 17: Change to 'heights h to provide
corrected
31) Page 11, line 19: I am not sure if all readers will know what is meant with an 'intensive quantity', maybe explain briefly in a footnote?
Ok, we add as a footnote: "bulk property which does not change when changing the size of the system"
We also noted that we did not explicitly mention the first step where we convert the relative intensive profile to an extensive one in the frist place (for scaling) before we convert it back to an intensive one. Which is very confusing.  We would like to change this by changing line 18 to "This profile, turned into a partial optical depth per layer by multiplying with the layer thickness, is scaled to match the total aerosol extinction from AERONET $\tau_{\rm aer}$. The profile is then converted back into an..."
32) Page 13, Equation 16: Rogue bracket or is something missing?
Opening bracket removed
33) Page 13, line 18: Gain is written in a strange font, on purpose (why?)? If not, please fix.

Changed to same font as AK everywhere

34) Page 13, line 21: 'produces'

corrected

35) Page 14, line 9: Add comma after fitting

corrected

36) Page 14, line 23: Change to 'AK matrices from the other errors.'

corrected

37) Page 15, line 1: 'the VMR(VMR)' – is that correct?

Yes, it describes the difference between  subscript "VMR" and "pcol"

38) Page 15, lines 16/17: Why would the vertical aerosol axtinction profile not be available?

Because the aerosol retrieval failed, or because it was judged to be a bad retrieval due to a  large rms w.r.t. measured and simulated dscd

39) Page 16, Figure 4 caption, last sentence: 'an ideal'

corrected

40) Page 16, line 6: Add comma after 'operator' – makes this sentence a bit easier to read.

corrected

41) Page 17, Equation 25: Should that be 3% instead of 0.3%?

yes, corrected

42) Page 17, line 8: Should either be 'error . . . is' or 'errors . . . are'

corrected

43) Page 17, line 14: Comma after retrieval

corrected

44) Page 17, line 15: 'contributions: a) smoothing error and b) .. error.

corrected

45) Page 18, line 1: Comma after (2017)

corrected

46) Page 18, line 3: Could use 'dependent' instead of 'not independent'.

corrected

47) Page 18, line 5: Delete 'it'. Comma after 'However'

corrected

48) Page 18, line 9: Delete 'the' before 'VLIDORT'.

corrected

49) Page 18, lines 10/11: Add commas after '(2017)' and 'the residual'

First coma added, second would be incorrect, we believe

50) Page 19, Figure 6: The two solid orange lines are hard to distinguish, could use dash or dash/dot for one of them.

We agree with the reviewer and will change one of the orange lines to a dashed orange line.

51) Page 19, Figure 6 caption: Change to 'a) The square . . .' and delete full stop after 'total'

corrected

52) Page 19, line 3: Better: 'errors for No2 and O4 calculated '

corrected

53) Page 19, line 5: Change to 'errors' and delete 'fairly'

corrected

54) Page 19, line 7: Delete 'relatively'

corrected

55) Page 19, line 8: Something is not right with this sentence & it doesn't make sense as it is written. Maybe delete 'to' or rephrase altogether.

Changed to two sentences: "The error in the vertical column is smaller than the errors in the VMR profile for almost all layers (Fig.6). This can be explained by an anti-correlation in different partial column errors indicated by the full error covariance matrix."

56) Page 20, Table 1, caption: The last sentence is a bit hard to read; would help to add a comma after included and it needs a 'with' after better.

We reformulated to:

"However, if the algorithm error according to Wang et al. (2017) is included, the remaining error due to the uncertainty in the aerosol profile is slightly better: 9.3\% instead of the 9.8% without $O_4$ retrieval."

57) Page 20, line 2: Add comma after 'In this section'

corrected

58) Page 20, line 4: Typo: 'approx.'

corrected

59) Page 21, line 10: Typo: 'Currently'

corrected

60) Page 23, line 1: I would rather say: 'Generally, a better . . .'

corrected

61) Page 24, lines 8-11 and Figure 10: Would be really interesting to get higher resolved surface measurements as well, otherwise a small increase might be hidden in the surface data set as well. The peak only shows up clearly in the individual measurements with sufficiently high temporal resolution. Similar peak also shows up on Aug 15 and one could argue to some degree even on 9 Sep and 22 Dec with a bit of a time shift. Any idea what causes it?

We think this NO2 enhancements might be transported from somewhere within the basis. Definitely, surface data with higher temporal resolution would allow us to do a more in-depth analysis on a day-to-day basis, as mentioned to referee 2. We think, however, that such a detailed study would divert from the main objectives of the paper, which is to describe the methods and quality of these data.

62) Page 24, line 28: Change to: 'This might have to do with the fact that during . . .' otherwise something seems to be missing from this sentence.

Yes, indeed. corrected

63) Page 24, lines 20-23 & Page 26, Figure 11: Could you add a brief discussion here on the nicely (amazingly?) constant offset between surface and MAX-DOAS data, also including the uncertainties of both data sets in that discussion. Would you say that this is predominantly caused by NO2 having strong emissions on the surface which are then just diluted over the vertical range which the MAX-DOAS measurements are covering?

Thank you for this comment. We added the following sentence at the end of the paragraph (Page 24, line 23). "Despite the fact that the offset in the curves for surface- and MAX-DOAS measurements appears to be nearly constant throughout the day, it would be interesting to investigate further how this offset varies in different seasons particularly when vertical mixing is not favoured".

64) Page 26, line 1: Change to: 'and certain trace gases. We . . . NO2 at one . . .'

corrected

65) Page 27, line 1: delete ' 's'

corrected

66) Page 27, line 4: Add something like 'Sincs this study, it has been . . ..'

We take this suggestion

67) Page 27, line 10: Add 'the' before 'NO2'.

corrected

68) Page 31-34, References: There seems to be some doubling up of information, please check through all the references for correct formatting.

Thank you for the note on this, we looked over the reference formatting, see also reply to referee 2 on this subject.

---

## Author Comment (AC3) · 7 Feb 2019

We would like to thank reviewer 3 for their comments. In the following, we respond to each of the comments individually. In order to facilitate the tracking, we use the following color coding:

Color coding:
      reviewer comment
      our answer
      proposed change in manuscript
* * *
General comments:
1/I think it would help the reader to include a map (that could be Figure no 1 of the manuscript) of the Mexico City area showing the locations of the different instruments (MAX-DOAS, AERONET, in-situ). Moreover, indicating the pointing directions of the MAX-DOAS instrument involved in this study could maybe also give some insights on the interpretation of the discrepancy between MAX-DOAS and in-situ NO2 surface concentration values. For instance, part of the underestimation of the in-situ values by the MAX-DOAS could be related to the fact that the MAX-DOAS instrument points towards a part of Mexico City which is less polluted than the location of the in-situ instrument.
In addition to the map, the location (latitude, longitude, altitude) of all the instruments should appear in the text.
The AERONET site and the in-situ measurement site are at the same position (give a nd take a few tens of meters) as the MAXDOAS station. The site for the ballon launch for the sounding is at the airport (a few kilometers to the north). However, since the position of the ballon quickly changes, we do not think that the actual launching position is very important. We will however state the coordinates in brackets below the first mentioning of the measurements. As for a map for the MAXDOAS instrument and its orientation, such a map is included in Arellano et al. 2016. We will refer to that in the manuscript specifically.

2/AERONET data are used as input in the retrieval but also as ancillary data for the sky conditions screening. Was there any attempt to compare the retrieved AODs with those from AERONET ? It can be a good check for the aerosol retrieval part of the profiling. Also related: it seems that the availability of AERONET observations has been used as a quality control (QC) flagging for the MAXDOAS retrievals. Was there any attempt to apply a QC flagging which is more specific to the MAX-DOAS retrievals, e.g. using parameters like DOF and the RMS of the differences between measured and calculated dSCDs ?
We first clarify which aeronet data is used in which way (a) and then answer the question about the filtering (b):
(a) We use the AERONET data of omega and g as input for the forward model (time interpolated). We do not attempt to retrieve those values. Further, the extrapolated (at the aerosol retrieval wavelength, and time interpolated) aod value is used as a-priori. As input for the NO2 retrieval, an interpolation between the retrieved aod and the nearest AERONET wavelength (time interpolated) was performed, see also answer to question "specific 2" below.  In case of failure of the aerosol retrieval (non-convergence, or a bad fit in terms of  the average absolute value of the difference in measured and simulated dscd in units of dscd error [sum(abs(DSCD_sim-DSCD_meas)/DSCD_meas)/number_of_elevation_angles] but no filtering on DOF because we designed the scaling of the Thikonov constrained in such a way to have a DOF of just above 1), an extrapolation of the two nearest (both to the long-wavelength side) AERONET values (time interpolated) was performed. Hence we do not see too much sense to compare to AERONET.

(b) In order to ensure a small forward model error it is important to have good estimations on the aerosol parameters g and omega. Without AERONET data available, we use an interpolation of the nearest available data in time. Hence the forward model error is expected to be smaller if AERONET data is available close in time. We also use the presence of AERONET data as a proxy for cloud free conditions. See also the answer to question 4a from reviewer 2.
Regarding the NO2 retrieval we do currently not use any filtering, not on RMS not on DOF. We looked at the distribution of DOF. We show a histogram in Fig. 1

[Figure]

Fig.1 Distribution of DOF for NO2 retrievals.

If we were to include a filter on DOF, we would likely choose a limit of 1.5, hence the impact had been rather small.

Specific comments:

1/Page 4, beginning of Section 2.3: It would be good to list the exact elevation angle values of a typical scan.
We will include this in the last line on page 4, just before mentioning the likely change in measurement sequence.

2/According to Section 3, it seems that aerosol profile retrieval is done in the UV range and then retrieved profiles are used as input for the NO2 profile retrieval in the visible range. Has a correction been applied to the retrieved extinction/AOD for taking into account for the wavelength dependence of the AOD/extinction ? If not, then this approximation should be included as an additional error source in Table 1.
We perform a linear interpolation at the NO2 retrieval wavelength using the value retrieved value in the UV and the closest AERONET value. We plan to use an O4 retrieval window closer to our NO2 retrieval window in the future. The error arising from this would be one contribution to the estimated algorithm error which is already in the table (The error in NO2 from errors in the aerosol profile). The easiest way to get this error contribution would be perhaps to use PANDORA

instruments (calibrated-direct sun measurements) collocated to AERONET sites. But up to now there are not yet sufficient coincident measurements in Mexico City

3/Page 6, lines 8-11: Maybe you could add a couple of sentences about the performance of MMF in these profile comparison exercises. Please note that in the meantime, Friess et al. is now published in AMTD.
Thank you for the note that Friess et al. Is now published. This will be added to the references. The inclusion of the changes in MMF is moved to an appendix (see comment to question 2b from reviewer 2), a note on the time performance is included.

4/Page 8, line 7: Is it IO or HONO (cf page 6, end of Sect. 4) ?
It is HONO, we corrected this and also added the missing reference to the list of references.

5/Page 8, Sect. 4.2: You should add a paragraph on the SCIATRAN RTM, which has been also used in past MAX-DOAS profiling studies (see e.g. Friess et al., AMTD, 2018).
We will add SCIATRAN RTM in the list of examples for radiative transfer models used as forward models for profile retrieval with MAXDOAS.

6/Page 10, line 1: what type of interpolation is done for the pressure, temperature profiles ?
We use simple linear interpolation. We know that this can be improved upon for P. However we expect the effect to be rather small since the grid for the T and P profiles is of a similar resolution than the internal retrieval grid.

7/Page 10, line 14: a correlation length of 500m is used. Did you perform sensitivity tests on this parameter in order to estimate its impact on the retrieved profiles and on the level of agreement with in-situ measurements ?
The correlation length was only used for the error calculation, not for the retrieval. For the retrieval, no Sa matrix was constructed, but a Thikonov constrained used.

8/Page 19, Figure 6b: A priori profile should be also included in this Figure in order to see how far the retrieved profile differs from the a priori one.
We will include the a-priori profile in the plot. Also, as response to reviewer 1, we change one of the orange line for easier distinction. A proposed new Figure 6 (in the manuscript) is reproduced here as Fig. 2 (we will adjust the axis labels and tick labels to a more readable font size)

[Figure]

Fig. 2: as Figure 6 in the manuscript but including the a-priori (right) and a better distinction of the curves (left).

9/Page 26, lines 5-10: According to the authors, a possible reason for the underestimation of the in-situ surface concentration by the MAX-DOAS is the fact that the MAX-DOAS instrument has a maximum sensitivity around 1km and less sensitivity close to the ground. This feature is quite unusual since normally lowest elevation angles have a higher weight in the retrieval due to higher AMFs, and therefore MAX-DOAS measurements close to the ground. Could the authors elaborate on that ? Another possible explanation for the discrepancy is that, related to the horizontal extent and the pointing direction of the MAX-DOAS measurements, both instruments probe different air masses. I think this point should be also added in the discussion.

The averaging kernel, for a typical AK see Fig.4(a), shows that we expect an underestimation at the surface. It peaks at around 1km. The main reason for the lower sensitivity is the measurement angle distribution (too few low elevation angles) and the rather huge dscd errors at these low elevation angles, see Fig.2. AK=(K^T Se-1 K+Sa-1)-1 (K^T Se-1 K). While Sa^-1 is constant, a full covariance matrix from model averages, Se-1, the measurement error and K, the Jacobian, mainly dependent on the aerosol content, are variable. Therefore, the AK is only an example.

We can actually estimate the slope and underestimation theoretically using a typical Averaging Kernel using the variability of NO2 in the Mixing layer described by the Sa. However, since Sa is only an estimation, we tried two Strategies a) either using the Sa calculated from profiles of the model run and another taken from the Literature Wang et al., 2017. The first one is used as constraint in the OET-retrieval and described in the Manuscript. The latter uses a 100% variability of the a priori an the diagonal and a 500m exponential correlation length for off-axis elements as in Wang et al. (2017).

The Slope between a retrieved quantity, either the total column or the average of some layers, is calculated, respectively, by applying an operator g=( 1 1 1 1 1 1 .... ) on the profile in units of partial columns or g6=(1/6, 1/6 ,1/6, 1/6, 1/6, 1/6,0,0,0....) in VMR. "g6" is the operator which calculates the average of the lowest 6 layers. To get the in situ value we apply g1=(1 0 0 0 0...) on the profile.

The linear relation between retrieved values (e.g. averages of 6 layers) and in situ values depend on the correlation between all layers and is theoretically described by the following expression:

$< g_6 AKV_{VMR}|SA_{VMR}| g_1>$ which assumes that the profile variability is described by a normal distribution P(x)=1/sqrt(Pi Det(SA)) Exp(-(x-xa)^t SA-1 (x-xa))  (Rodgers 2000) of the Variability.

The theoretically calculated slope can be compared to the experimental obtained slope (Fig. 8) and so the Sa- matrix can be tested for, how plausible the estimation was.

In Fig. 3 here, we show the slopes (y-axis of Maxdoas v.s. insitu for the average of a different numbers of the lowest layers indicated by the x-axis (just as in Fig. 8 in the paper).

[Figure]

Fig3: As Figure 8 in the manuscript, just with a few more lines with respect to different filters related with the aeronet data and we added the two theoretical calculations shown by dots, green dots using the SA described in this work and the pink dots using the SA-Matrix described by Wang et al, 2017. Details see text.

The points are calculated theoretically either using the SA we constructed from the ensemble of modeled profiles (green points, Slope_arr) or taken from Wang et al., 2017 (pink points, Slope_arr_aer).

The graph explains even quantitatively the underestimation and no other arguments are needed. Just for the average of very few layers, there seems to be a discrepancy, which might indicate that the SA-Matrix do not describe the variability and correlation of the lowest level correctly.

We learned a lot by this exercise, but we would prefer not to complicate the manuscript too much and suggest just to add

"and variability Sa" behind the averaging kernel and maybe add as well the two calculated values

10/Acknowledgements: Depending on the conditions of use, the sources of ancillary data included in your study should be acknowledged here.

Thank you for pointing this out, we will include appropriate acknowledgements.

Technical corrections:
1/Page 1, line 1: '. . .to retrieve profiles. . .' -> '. . .to retrieve vertical profiles. . .'
corrected
2/Page 1, line 10: '. . .at the Universidad Nacional Autónoma de México (UNAM) cam-pus.'
corrected
3/Page 1, line 20: '. . .The Multi-AXis Differential Optical Absorption Spectroscopy

(MAX-DOAS) technique. . ..'

corrected

4/Page 2, line 30: '. . .at the Universidad Nacional Autónoma de México (UNAM).'

corrected

5/Page 6, line 5: 'Sect.4.1' -> 'Sect. 4.1'. This typo should be corrected throughout the manuscript; similar corrections also needed for 'Fig.' and 'Eq.'.

Sect, Fig. And Eq. checked for space behind.

6/Page 6, line 9: 'CINDI2' -> 'CINDI-2'

corrected

7/Page 10, line 14: '. . .Eq. 4:'; same on Page 11, line 4

corrected

8/Page 11, line 20: 'extincion' -> 'extinction'

corrected

9/Page 16, legend of Fig. 4: (b,right)' -> '(b, right)'; should be also corrected for (a, left).

corrected

10/Page 19, legend of Fig. 6: 'total error'

corrected

11/Page 19, line 3: 'algoritm' -> 'algorithm'

corrected

12/Page 20, line 4: 'aprox.' -> 'approx.'

corrected

13/Page 21, line 11: 'Curretly' -> 'Currently'

corrected

14/Page 24, end of line 23: A reference could be added here.

We will add García-Franco et al. 2018

---

## Author Response (AR2)

This file is organized as follows:

- (1) Answer to reviewer 1
- (2) Answer to reviewer 2
- (3) Answer to reviewer 3
- (4) Marked-up version (against last submitted version)

Each answer to a reviewer (1 - 3) has the following color coding:

reviewer comment our answer change in manuscript addition w.r.t the last version of answer

We included now in every answer the change in the text, also for the typos. However, since the vast majority of typos was already corrected in the last submission, they do not show up in the new mark-up file.

We also checked for the reference to equations in the text and found this sometimes missing. In these cases, we slightly changed the formulation so that each equation is explicitly referred to in the text. These changes are also highlighted in the marked up version (4) below.

**Answer to Reviewer 1**

We would like to thank the reviewer for his comments. Since we received a comment from reviewer two before we received a comment from reviewer one, we are sometimes referring in this answer to the answer to reviewer 2.

Below, we comment on the first reviewers specific comments. For easier reference, we added a number to each comment. We use the following color coding:

Review of "NO2 vertical profiles and column densities from MAX-DOAS measurements in Mexico City" by Friedrich et al.

This manuscript discusses a newly developed profile retrieval code - the Mexican Maxdoas Fit (MMF). Note: The first author's initials match the acronym, nicely done! The retrieval code consists of 2 parts, 1) an aerosol retrieval and 2) a trace gas retrieval using the previously retrieved aerosol profiles. This code is then used on 19 months of MAX-DOAS data measured at a location in Mexico City and the results are discussed. A comprehensive error analysis (which is great to see!) is also included in the manuscript.

It certainly is interesting to look at the complete 19 months NO2 data set (e.g. see the discussed averages of the diurnal variation) but my guess is the more interesting studies (specially from an environmental view point) can be done by looking at individual days and using the right ancillary data to understand the NO2 variability and what causes the observed peaks.

The reviewer takes up this point again in his question 61. Also reviewer 2 touches on this in his question 4b. As we mention in the answer to those questions, we think that such detailed studies of individual days would divert from the main objectives of the paper which is to describe the methods and quality of these data. As is mentioned in Sect. 7 of the manuscript, data from all 4 stations is currently analyzed: "Although results are shown only from one instrument located in the southern part of the city (UNAM), several years of data are available from three other stations at different locations within the metropolitan area (Acatlán, Vallejo and Cuautitlán) that are being analyzed and used for studying the spatial and temporal variability of NO 2 in conjunction of several satellite products"

Overall, the manuscript is well structured and the figures and table are clear and straight forward to understand. However, in some places (e.g. in Section 5, Error analysis) the text can be somewhat difficult to follow, and the manuscript could gain from having another go at streamlining the text a bit more and simplifying the structure of some of the more complicated sentences.

We agree with the referee that the text was a bit hard to follow in places and we hope that we could improve this by following both his and the other reviewers advices (see below and in the answers to the other reviewers questions) and suggestions for improvement. Major work has been carried out to reformulate Sect. 5. Complicated sentences have been split and all equations are now specifically referred to in the text. Also, following advice of reviewer 2, all references to Table 1 are now made as early as possible and with a specific line number. We name a few example of reformulations below. Please check the marked up version at the end of this document for a complete overview of changes.

The Gain matrix, as defined by Eq. 17, describes the change in retrieved x when measurement y changes <del>and</del>. It can hence be used to map the uncertainty in measurement space , to a state-vector uncertainty.

How a change in the true atmospheric state vector x\_true is translated into changes in the retrieved state vector x is described by the partial column Averaging Kernel matrix AK\_pcol The partial column Averaging Kernel matrix AK\_pcol describes how a change in the true atmospheric state vector x\_true is translated into changes in the retrieved state vector x

The Averaging Kernel of partial column and total column AK\_tot describe how the retrieved solution profile depends on the real atmosphere x\_true <del>and</del>. They have to be taken into account if the profile and the column is used.

The Averaging Kernel matrix is in general asymmetrical and the. The columns are representing response functions related to a perturbation in a certain layer. The rows of the matrix represent while the sensitivity of the NO2 partial column of a certain layer (or concentration in a certain layer) to all the different true partial columns or concentrations. are expressed by the rows of the matrix.

How the Averaging Kernel changes under transformation of the units is expressed by two matrixmultiplications, one with the diagonal matrix containing the partial air column in its diagonal-U\_aircols from one side and its inverse-U^-1\_aircols from the other side.

The change of Averaging Kernel under transformation of the units is expressed by two matrix multiplications: One by multiplying the diagonal matrix containing the partial air column in its diagonal U\_aircols from the right with the Averaging Kernel; the other by multiplying the inverse of the diagonal matrix U^-1\_aircols from the left side with the Averaging Kernel.

For the calculation we We have assumed a constant sensitivity AK and a constant a priori covariance matrix Sa for the calculation.

The AK of the trace gas profile indeed depends strongly on the aerosol profile <del>and even</del>. It also depends slightly on the trace gas profile itself <del>and the</del>. The Sa covariance matrix of the aerosol extinction profile should be given for each hour as the a priori profile varies with time of day.

1) Page 2, line 16: Replace 'giving' with 'with'

corrected giving with

2) Page 2, line 17: Section 2 should be Sect. 2 for consistency, check whole manuscript checked in whole manuscript, thank you! Section 2 Sect.2

3) Page 2, line 19: '(constituting the forward model)' - What exactly does this mean? This means that the forward model in our case is a radiative transfer code and that we talk about radiative transfer codes here, because our forward model is a radiative transfer model, more specifically, VLIDORT.

Changed to: (constituting the forward model in MMF)

4) Page 2, line 30: UNAM – can you please spell this out once done at first appearance now, i.e. at the beginning of Sect. 2 ...at the Universidad Nacional Autónoma de México (UNAM)

5) Page 3, Figure 1, caption. Nice overview figure. For completeness, can you please also include a brief description of the yellow and red box in the caption.

Yes. As response to reviewer 2, we also made small adjustments to the figure, see answer to referee 2. The new caption is describing that version of the figure. We added to the caption:

"The yellow boxes represent the forward modeling steps. The red boxes are the inversion steps, using Thikonov regularization for aerosol retrieval and optical estimation (OE) for tracegas retrieval."

As a response to (23) below, we also now added "& rates of change" in the orange and green boxes before the yellow "VLIDORT box".

6) Page 3, line 3: Replace 'large' with 'long'

corrected large long

7) Page 4, line 8: Typo: 'receiving'

corrected receiving

8) Page 5, line 4: Typo: 'an average'

corrected an average

9) Page 5, lines 16-25: Why was O4 not retrieved using the same wavelength interval as NO2? The much older O4 XS from Hermans et al. 1999 was used for the O4 retrieval, why not Thalman and Volkamer (2013)?

Thalman, R. and R. Volkamer, Temperature dependent absorption cross-sections of O2-O2 collision pairs between 340 and 630 nm and at atmospherically relevant pressure., Phys. Chem. Chem. Phys., 15(37), 15371–81, doi:10.1039/c3cp50968k, 2013. The windows for NO2 and O4 dscd fitting are chosen to enclose pronounced absorption lines for the species in question and are widely used windows. We agree with the referee that there are O4 windows that are closer to the chosen NO2 window (405 -- 465 nm) than our choice (336 -- 390 nm) and that it had been a better choice to use one of those (e.g. 450 -- 520 nm). However, the difference in middle wavelength had only been 22 nm less (i.e. +50 instead of -72). We would also like to refer to our answer to reviewer 3 question 2 regarding correction for aod. Regarding the

choice of cross-sections, there was no specific reason for the choice of cross-section. We would like to refer the reviewer 1 to answer 1b to the mayor comments from reviewer 2, where we also include a test for changing the retrieval settings. Our main finding is that the effect of changing the cross-sections is small. Hence we do not intent to do any changes to the manuscript.

**10) Page 5, line 26: Would it be possible to say something briefly here about how the errors were determined?**

The dscd error is calculated directly within qdoas. We use this error as dscd error without any modification or addition. We refer the reviewer to pages 28 -- 29 ("Errors on Slant Column Densities) of the qdoas manual (http://uv-vis.aeronomie.be/software/QDOAS/QDOAS\_manual.pdf) for details on the dscd error calculation within qdoas. We include the reference in the text: The dSCD errors are the QDOAS calculated errors, a function of the fit residuals and the degree of freedom, see Dankaert et al. (2013) for details.

11) Page 7: The authors explain that the retrieval code was recently updated from using

the Gauss-Newton scheme to the more stable Levenberg Marquardt iteration scheme. However, this is not really relevant for the work presented here and seems to unnecessarily complicate the discussion. Unless there is a compelling reason to keep this information, I suggest to drop the relevant equations and just briefly mention in a couple of sentences (or one paragraph) that the retrieval code has been updated and how. It would also be better to have all the variables explained straight after Equation (1) and not further down the page.

We fully agree with the referee. Referee 2 (first report received) had a similar comment. We moved all explanations regarding changes to the code into an appendix. This also leads to all symbols in Eq. 1 being defined right after its appearance. For details on the changes, please see the answer 2b to the major comments section from referee report from reviewer 2 where we list in detail all the changes made to the manuscript.

12) Page 7, line 6: Change to 'non-linear'

corrected non-linear

13) Page 7, line 12: Change to 'dimension which is the number of telescope'

done *#angles*-dimension which is the number of telescope viewing angles

14) Page 7, line 17: Change to: 'equal to 1'

corrected equal to 1

15) Page 7, line 23: Change to: 'for the trace gas'

included "the" for the trace gas

16) Page 8, line 8: Change to: 'with the LM iteration scheme.'

included "the" with the LM iteration scheme

17) Page 8, line 10: Change to: 'algorithms. For example, there are'

Added "For example" For example, there are

18) Page 8, line 20: Typo: 'high speed'

corrected high speed

19) Page 8, line 21: 'instead of the 2x the number of layers calls'

corrected instead of the 2x the number of layers calls

20) Page 8, line 23: Jacobians always with capital J, also on Page 12 & 13 now capitalized everywhere, see the marked up version for details.

21) Page 8, line 23/32 and footnote: Why not refer straight to LIDORT if only that part if used anyway?

VLIDORT and LIDORT are actually different code packages with different version numbers. Since it might be that they, of course by accident, include different "features" (i.e. bugs), we think that it is more accurate to state exactly which code and which version was used. Hence we leave VLIDORT instead of changing it to LIDORT.

22) Page 8, line 32: Maybe replace with 'For each simulated atmospheric layer, ' corrected For each simulated atmospheric layer,

23) Page 9, lines 9-14: It is not quite clear to me how the rate of change is represented in Figure 1, can you please explain . . . or I might have misunderstood?

This was perhaps not clear. We did not refer specifically to the rate of change. We removed the sentence. The part of the input calculation is inclosed in Fig. 1 in the light blue box.

However, we also realized that the rate of change as layer input was indeed missing in the diagram. We added "& rates of change" in the orange and green box before each of the yellow VLIDORT boxes.

24) Page 9, line 13: Should be either 'enclosed' or 'included'? enclosed

25) Page 9, line 21: Replace 'is' with 'are'

corrected is are

26) Page 11, line 2: Comma needed after Qray

corrected Q Ray,

27) Page 11, line 14: 'are assumed to be constant in all layers.'

corrected are assumed to be constant in all layers.

28) Page 11, line 15: Replace 'are' with 'is'

corrected are is

29) Page 11, line 16: Change to 'density profiles in arbitrary units from . . .'

corrected density profiles in arbitrary units from

30) Page 11, line 17: Change to 'heights h to provide

corrected heights h to provide

31) Page 11, line 19: I am not sure if all readers will know what is meant with an 'intensive quantity', maybe explain briefly in a footnote?

Ok, we add as a footnote: "bulk property which does not change when changing the size of the system"

We also noted that we did not explicitly mention the first step where we convert the relative intensive profile to an extensive one in the frist place (for scaling) before we convert it back to an intensive one. Which is very confusing. We would like to change this by changing line 18 to "This profile, turned into a partial optical depth per layer by multiplying with the layer thickness, is scaled to match the total aerosol extinction from AERONET \$\tau\_{\mathbf{k}} aer}\$. The profile is then converted back into an..."

32) Page 13, Equation 16: Rogue bracket or is something missing? Opening bracket removed:

$$dSCD = \underline{\logln} \left( \frac{I_0^{\alpha} \cdot I_g^z}{I_g^{\alpha} \cdot I_g^z} \right) / \underline{(} \xi_{\lambda}$$

33) Page 13, line 18: Gain is written in a strange font, on purpose (why?)? If not, please fix.

Changed to same font as AK everywhere, see marked up version, e.g. in Eq. 17:

GainGain :

34) Page 13, line 21: 'produces'

corrected produces

35) Page 14, line 9: Add comma after fitting

corrected least square fitting, : the

36) Page 14, line 23: Change to 'AK matrices from the other errors.'

corrected AK matrices from the other errors

37) Page 15, line 1: 'the VMR(VMR)' – is that correct?

Yes, it describes the difference between subscript "VMR" and "pcol"

38) Page 15, lines 16/17: Why would the vertical aerosol axtinction profile not be available? Because the aerosol retrieval failed, or because it was judged to be a bad retrieval due to a large rms w.r.t. measured and simulated dscd

39) Page 16, Figure 4 caption, last sentence: 'an ideal'

corrected an ideal

40) Page 16, line 6: Add comma after 'operator' – makes this sentence a bit easier to read. corrected operator,

41) Page 17, Equation 25: Should that be 3% instead of 0.3%?

yes, corrected -0.3% 3%

42) Page 17, line 8: Should either be 'error . . . is' or 'errors . . . are'

corrected error ..... is

43) Page 17, line 14: Comma after retrieval

corrected retrieval,

44) Page 17, line 15: 'contributions: a) smoothing error and b) .. error.

Corrected contributions: : a) smoothing error

45) Page 18, line 1: Comma after (2017)

corrected Wang et al. (2017), the

46) Page 18, line 3: Could use 'dependent' instead of 'not independent'.

corrected not independent dependent

47) Page 18, line 5: Delete 'it'. Comma after 'However'

corrected which-it is not the However, errors...

48) Page 18, line 9: Delete 'the' before 'VLIDORT'.

Corrected by the VLIDORT

49) Page 18, lines 10/11: Add commas after '(2017)' and 'the residual'

First coma added, second would be incorrect, we believe

Following Wang et al. (2017), we

50) Page 19, Figure 6: The two solid orange lines are hard to distinguish, could use dash or dash/dot for one of them.

We agree with the reviewer and will change one of the orange lines to a dashed orange line. **New color scheme is:**

🛶 Variability

- Smoothing error
- Forward model error
- Measurement noise
- Spectroscopic
- --- Aerosol smoothing error
- Aerosol noise error
- --- Aerosol error no O4retrieval
- --- Aerosol forward error
- Total error

51) Page 19, Figure 6 caption: Change to 'a) The square . . .' and delete full stop after 'total' corrected a) The square root (total– error and variability)

52) Page 19, line 3: Better: 'errors for No2 and O4 calculated '

corrected errors for No2 and O4 calculated

53) Page 19, line 5: Change to 'errors' and delete 'fairly'

corrected These results are fairly errors are similar

54) Page 19, line 7: Delete 'relatively'

corrected the vertical column is relatively smaller than

55) Page 19, line 8: Something is not right with this sentence & it doesn't make sense as it is written. Maybe delete 'to' or rephrase altogether.

Changed to two sentences: "The error in the vertical column is smaller than the errors in the VMR profile for almost all layers (Fig.6). This can be explained by an anti-correlation in different partial column errors indicated by the full error covariance matrix."

56) Page 20, Table 1, caption: The last sentence is a bit hard to read; would help to add a comma after included and it needs a 'with' after better.

We reformulated to:

"However, if the algorithm error according to Wang et al. (2017) is included, the remaining error due to the uncertainty in the aerosol profile is slightly better: 9.3% instead of the 9.8% without  $O_4$  retrieval."

57) Page 20, line 2: Add comma after 'In this section'

corrected In this section, we present

58) Page 20, line 4: Typo: 'approx.'

corrected <del>aprox</del> approx.

59) Page 21, line 10: Typo: 'Currently'

corrected Currelty Currently

60) Page 23, line 1: I would rather say: 'Generally, a better . . .'

corrected Generally, a <del>good</del> better

61) Page 24, lines 8-11 and Figure 10: Would be really interesting to get higher resolved surface measurements as well, otherwise a small increase might be hidden in the surface data set as well. The peak only shows up clearly in the individual measurements with sufficiently high temporal resolution. Similar peak also shows up on Aug 15 and one could argue to some degree even on 9 Sep and 22 Dec with a bit of a time shift. Any idea what causes it?

We think this NO2 enhancements might be transported from somewhere within the basis. Definitely, surface data with higher temporal resolution would allow us to do a more in-depth analysis on a day-to-day basis, as mentioned to referee 2. We think, however, that such a detailed study would divert from the main objectives of the paper, which is to describe the methods and quality of these data. However, we intent to identify interesting days to investigate the data in more depth in future projects.

62) Page 24, line 28: Change to: 'This might have to do with the fact that during . . .' otherwise something seems to be missing from this sentence.

Yes, indeed. corrected This might have to do with the fact that during

63) Page 24, lines 20-23 & Page 26, Figure 11: Could you add a brief discussion here on the nicely (amazingly?) constant offset between surface and MAX-DOAS data, also including the uncertainties of both data sets in that discussion. Would you say that this is predominantly caused by NO2 having strong emissions on the surface which are then just diluted over the vertical range which the MAX-DOAS measurements are covering?

Thank you for this comment. We added the following sentence at the end of the paragraph (Page 24, line 23). "Despite the fact that the offset in the curves for surface- and MAX-DOAS measurements appears to be nearly constant throughout the day, it would be interesting to investigate further how this offset varies in different seasons particularly when vertical mixing is not favoured".

64) Page 26, line 1: Change to: 'and certain trace gases. We . . . NO2 at one . . .'

corrected -target- certain trace

65) Page 27, line 1: delete ' 's'

66) Page 27, line 4: Add something like 'Sincs this study, it has been . . ..'

We take this suggestion Since this study, it has been

67) Page 27, line 10: Add 'the' before 'NO2'.

corrected as for the NO2 retrieval

68) Page 31-34, References: There seems to be some doubling up of information, please check through all the references for correct formatting.

Thank you for the note on this, we looked over the reference formatting, see also reply to referee 2 on this subject. Unfortunately, there were problems with the mark-up in the reference section. There is no mark-up in the reference. We chose to stick to the doi link, unless there was an issue with the doi link, as e.g. for the first reference. In these cases, we used the provided link.

**Answer to Reviewer 2**

We thank the referee for his very careful review, and his constructive suggestions. In the following, we answer his specific questions.

W.r.t the answer given in the comments, we slightly enhanced the new Appendix A and updated this here in this answer. W.r.t. the difference PDF: The updated references are not marked-up by color.

1) Regarding the dSCD retrieval and Section 3:

a) The authors use a zenith measurement prior to the scan to analyze the scan. If the upper atmospheric contribution to the dSCDs changes during the scan this can lead to signal in the measurements which is from the upper atmosphere being falsely attributed to lower altitudes especially in low eastward elevation angles at the end of the scan. The effect would be expected to lead to lower VCDs in the morning and higher VCDs in the evening, especially in winter. Because of the short 7 minute scan time this effect would likely only be significant at twilight. Does the instrument acquire data at twilight which are included in the analysis? Can the authors bound the impact of such an effect and compare it to the magnitude of their error budget?

Currently we are not considering any data measured at twilight in the analysis and hence the effect of geometry should be small within a 7 min time window. The difference for the zenith measurements is likely to reflect the difference in atmospheric conditions over 7 minutes. If these differences are huge, one of the assumptions made in the concept of MAXDOAS, same conditions for measurements in all off-axis directions, is violated and the scan in question most likely not suitable for profile retrieval. A way of estimating the error from using the same reference would be to look at the difference between two zenith dscds of two consecutive scans using the noon zenith sky as reference. We made such a test and found that typically, these differences are of the order of several 1041. molec2/cm5. This is smaller/ of the order of the typical fitting errors for O4 of typically a couple 10^42.

We added the following sentence to the manuscript in Sect. 3:

Using a single zenith reference before the scan instead of a linear interpolation of zenith measurements before and after the scan, makes correct measurements during twilight problematic: since upper atmospheric contributions to the slant column density change rapidly during twilight, stratospheric NO2 could falsely be attributed to the troposphere. Hence, in the study presented here, no twilight measurements are included.

b) For the fitting setting in the retrieval the authors use older cross-sections where newer cross-sections for the same gases are increasingly standard in the community e.g. (Damadeo et al., 2013; Peters et al., 2017). For O3 they use (Burrows et al., 1999) rather than (Bogumil et al., 2003) or (Serdyuchenko et al., 2014) and for O4 they use (Hermans et al., 1999) rather than (Thalman & Volkamer, 2013). Was there any particular reason for these choices? Is it based on Orphal, 2002 cited later?

There was no particular reason for using the chosen cross-sections, they were chosen because they are quite standardly used for retrievals. At the moment, we are re-running all our dataset using fitting settings

as in Peters et al. (2017). A comparison among our chosen settings and the Peters et al., (2017) settings is shown in Fig.1 (data for one scan sequence).

a) Our retrieval settings.

b) Peters et al., 2017 retrieval settings.

c) a and b plotted together

d) Difference (a-b) with the error bars:  $\sqrt{\varepsilon_a^2 + \varepsilon_b^2}$

From this graphs it is evident that the difference arising from the use of different cross-sections is small compared to the typical fitting error in the DOAS retrieval.

Fig1: Comparison of results for one scan using retrieval settings as used in the manuscript with results using retrieval settings as in Peters et al. (2017)

Since the effect is small, we do not intent to change the retrieval settings for this analysis. However, we will follow the recommendation of the reviewer for future studies.

2) Regarding the profile retrieval method description and Section 4

**a)** At present the aerosol and trace gas inversions in Fig. 1 are presented as the same, whereas the former uses Tikhonov regularization and the latter optimal estimation. This should be reflected in the figure as it is in the text.

We agree with the reviewer and would like to make the following change in Figure 1. We include a suggested Figure to replace Figure 1 in the manuscript here as Fig. 2.

\* Replace the first red box by: "Inversion using Thikonov Regularization"

\* Add on the right hand side of that box a box with "L1 Scaling fator" and connect this box with a left-ward pointing arrow to the red box

- \* Replace the second red box by "Inversion using OE"
- \* Add on the right hand side of this box a box with "Sa from simulations"

Also, reviewer 1 has pointed out a missing "rates of change" as input quantities, we also add this to the orange and green box that leads to the yellow VLIDORT boxes.